# IMPROVED FUNCTION-SPACE VARIATIONAL INFERENCE WITH INFORMATIVE PRIORS

## ABSTRACT

Function space variational inference allows Bayesian neural network (BNN) to introduce the prior distribution on the function space directly. Moreover, Recent linear approximation scheme for KL divergence between two random functions, has presented the tractable training objective and thus facilitates imposing the function space prior on BNNs. On the other hand, despite of its tractability, the existing inference suffers from the interpretability issue because the this function space prior is obtained by mapping the pre-defined weight-space prior to the function output via the complex neural network, and thus seems to be less interpretable. Alternatively, thought the uniform function space prior, that imposes a zero mean prior on the function space to encourage the model to be uncertain for out-of-training set, has been considered, this prior can introduce unnecessary uncertainty into the function outputs of the training datasets. Thus, this can cause the trade-off between the uncertainty estimation performances on the in-training and out-of-training sets.

In this work, we aim at refining the function space variational inference to handle the mentioned issue. To this end, we first reconsider the role of the function space prior in view of Bayesian Model Averaging (BMA), and then propose a refined function space prior via empirical Bayes and variational function space distribution that can sample the useful predictive functions in sense of BMA.

## 1 INTRODUCTION

Function space Bayesian neural networks (BNNs) (Sun et al., 2019) have gained significant attention within the Bayesian deep learning community, primarily due to their fundamental goal of directly assigning prior distributions to the outputs of neural networks. This approach stands in contrast to weight-space Bayesian neural networks, where interpreting prior distributions or incorporating prior knowledge can be intricate and less intuitive. In function space BNNs, it becomes possible to establish a prior distribution that directly embodies the inductive biases expected from neural networks, offering a more intuitive and user-friendly approach.

Despite their promising theoretical advantages, the practical implementation of function space BNNs often presents considerable challenges, mainly in their posterior inference procedures. A standard approach for the posterior inference of a function space BNN is variational inference (Sun et al., 2019), where one must compute the function-space KL divergence between stochastic processes. Unlike the KL divergences in weight-space distributions, the function space KL divergences do not admit easy-to-use Monte-Carlo estimators, so they are typically approximated with KL divergences between function values evaluated at a finite number of inputs. To partially alleviate these challenges, Rudner et al. (2022) introduced the concept of tractable function-space BNNs, where function space priors are formulated as mappings from weight-space priors. This formulation facilitates the estimation of function space KL divergences through linearization, offering a more practical approach to posterior inference in function space BNNs.

However, although the tractable function-space BNNs can be conveniently trained due its tractable inference, it has still some issue of constructing the reasonable prior distribution on the function space. As the function space prior is obtained by evaluating the linearized neural network, using the pre-defined weight-space prior as the weight parameters, on the additional inputs, the function space prior could be significantly different depending on the specification of the weight space prior and the

additional inputs. Thus, the function-space prior through the mapping would be less interpretable. Alternatively, the interpretable uniform function space prior, that imposes a zero mean prior on the function space to encourage the model to be uncertain on the out-of-training set, can be considered. However, this prior inevitably introduces unnecessary uncertainty into the function outputs of the training datasets, and thus can make models more uncertain than they need to be.

In this work, we aim at improving function-space BNNs by presenting the informative function space prior. To this end, we first reconsider the role of the function space prior in the sense of Bayesian model Averaging (BMA) and note that it would be a reasonable prior to improve the uncertainty estimation if the function space prior encourages the sample functions of the posterior distribution to have a certain predictive probability as well as have the varying degree of disagreements according to the status of predictive inputs. Then, we build an informative function space prior by using the empirical Bayes approach along with the parameters of hidden features and the last-layer weight parameters, which are obtained iterations during early stages of training. We also address another challenge of function-space BNNs that requires additional data for computing the KL divergences in function space. Unlike the previous works assuming the presence of external datasets or simply injecting noises to training data, we consider an adversarial hidden feature to construct the function space prior from using only training set. Finally, we refine the variational function space distribution to encourage the sample functions to have the varying degree of disagreements according to the status of the predictive inputs. Our contribution can be summarised as:

- Motivated by BMA, we propose an informative function space prior and the variational distribution over function space that can improve the uncertainty estimation of the BNNs.
- We consider the adversarial hidden features to construct the function space prior using only the training dataset without the need of the additional input.
- We demonstrate that the vision transformer trained by the proposed inference, outperforms other baseline inference methods on CIFAR 100, showing its effectiveness on the large-scale model.

## 2    BACKGROUND.

**Notations.**    In this work, we focus on Bayesian neural network (BNN) for supervised learning task. Let $\mathcal{X} \subset \mathbb{R}^D$ and $\mathcal{Y} \subset \mathbb{R}^Q$ be the space of dataset input and output, and $f : \mathcal{X} \times R^P \to \mathcal{Y}$ be a BNN that takes the input $\cdot \in \mathcal{X}$ and the random weight parameters $\Theta \in R^P$, following prior distribution $\Theta \sim p(\Theta)$, and produces the random output $f(\cdot, \Theta) \in \mathcal{Y}$. Without loss of generality, we assume that $f$ consists of $L$ layers, i.e., $f(\cdot, \Theta) = \left( f^{(L)} \circ \cdots \circ f^{(2)} \circ f^{(1)} \right)(\cdot)$, with $\Theta = \{\Theta^{(l)}\}_{l=1}^L$, where $\Theta^{(l)}$ denotes the $l$-th layer random weight parameters.

**Variational Inference for BNN.**    For given dataset $\mathcal{D} = \{(x_n, y_n)\}_{n=1}^N$ with input $x_n \in \mathcal{X}$ and $y_n \in \mathcal{Y}$, training a BNN requires computing the posterior distribution $p(\Theta|\mathcal{D}) = p(\Theta, \mathcal{D})/p(\mathcal{D})$ by Bayes theorem. However, since the posterior distribution of the weight parameters $p(\Theta|\mathcal{D})$ are not tractable in general, approximate Bayesian inference method have been used to train BNNs. Among the inferences, Variational inference (VI) introduces the variational distribution $q(\Theta)$ to approximate the posterior distribution $p(\Theta|D)$, and then updates the parameters of $q(\Theta)$ by maximizing the evidence lower bound $\mathcal{L}_{\text{vi}}(\Theta)$ defined as follows:

$$\mathcal{L}_{\text{vi}}(\Theta) := \mathrm{E}_{\Theta \sim q(\Theta)} \left[ \sum_{n=1}^N \log p\big(y_n | f(x_n, \Theta)\big) \right] - \lambda \, \mathrm{KL}\big(q(\Theta) \,\|\, p(\Theta)\big), \tag{1}$$

where $\lambda$[1] denotes the hyperparameter that controls the amount of the regularisation of the KL divergence. In general, variational distribution $q(\Theta)$ and prior distribution $p(\Theta)$ are assumed to be Gaussian distribution because Gaussian distribution allows the random parameter $\Theta$ to be differentiable by reparametrization trick and the corresponding KL term in Eq. (1) to be tractable.

**Function-Space Variational Inference for BNN.**    Function Space BNNs incorporates the inductive bias into the model output directly, and thus introduces the the prior distribution on the function space

---

[1] Setting $\lambda < 1$ leads a cold posterior effect, which has known to improve the uncertainty estimation of the BNNs (Wenzel et al., 2020). In this work, we also employ the cold posterior effect for all inference methods.

$\mathcal{Y}$. Let $p(f(\cdot))$ be the prior distribution of the model output $f(\cdot)$ and $q(f(\cdot))$ be the corresponding variational distribution. Then, function space BNNs are trained to maximize the ELBO $\mathcal{L}_{\text{fvi}}(\Theta)$:

$$\mathcal{L}_{\text{fvi}}(\Theta) := \mathrm{E}_{f\sim q(f(\cdot))}\left[\sum_{n=1}^{N}\log p\big(y_n|f(x_n)\big)\right] - \lambda\,\mathrm{KL}\big(q(f(\cdot))\,||\,p(f(\cdot))\big), \qquad (2)$$

where the KL divergence between two stochastic processes can be defined (Sun et al., 2019):

$$\mathrm{KL}\big(q(f(\cdot))\,||\,p(f(\cdot))\big) = \sup_{X_I:|I|<\infty}\mathrm{KL}\big(q(f(X_I))\,||\,p(f(X_I))\big). \qquad (3)$$

In words, the KL divergence between two stochastic processes are the supremum of the KL divergence between finite-dimensional distributions evaluated at a finite set $X_I := \{x_i \in \mathcal{X}; i \in I\}$, where $f(X_I) := \{f(x) \mid x \in X_I\}$ and similar for $q(X_I)$. In practice, evaluating the supremum is intractable, and it is typically approximated with a heuristically chosen finite evalution set $X_I$. Throughout the paper, following the convention, we refer to the set $X_I$ being used for the approximtion of the KL divergence as *context set*. Even with the approximation with a context set, depending on the choice of the variational distribution $q(f(\cdot))$, the KL term $\mathrm{KL}\big(q(f(X_I))\,||\,p(f(X_I))\big)$ in Eq. (3) may not admit a closed-form expression, in which case optimizing it involves the additional technique for gradient estimation (Sun et al., 2019).

**Tractable Function-Space Variational Inference for BNN.** Rather than directly eliciting a prior distribution $p(f(\cdot))$, one can first choose a weight space prior $p(\Theta)$ and then define the function space prior $p(f(\cdot,\Theta))$ as a induced distribution $p(f(\cdot,\Theta)) := \int_{\mathbb{R}^P}\delta_{\Theta}(\Theta')f(\cdot,\Theta')p(\Theta')d\Theta'$. Based on this prior, Rudner et al. (2022) proposed a tractable variational inference method for function-space BNNs, where the outputs of BNNs are linearized with respect to the weights to make the computation of KL term in Eq. (3) tractable. More specifically, let $p(\Theta) = \mathcal{N}(\Theta; \mu_w, \Sigma_w)$. The lineariztion of the output of a BNN indexed by $\Theta$ is,

$$f(\cdot,\Theta) \approx f_{\text{lin}}(\cdot,\Theta) := f(\cdot,\mu_w) + J^f_{\mu_w}(\cdot)(\Theta-\mu_w) \qquad (4)$$

where $J^f_{\mu_w}(\cdot) := [\frac{\partial f}{\partial\Theta}]_{\Theta=\mu_w} \in \mathbb{R}^{Q\times P}$ denotes the Jacobin matrix obtained by differentiating the function $f$ with respect to the mean parameter $\mu_w$. Then, one can easily see that the linearized random function $f_{\text{lin}}(\cdot,\Theta)$ follows the Gaussian distribution, defined as:

$$p(f_{\text{lin}}(\cdot,\Theta)) = \mathcal{N}\Big(f_{\text{lin}}(\cdot)\,;\,\mu_f(\cdot)\,,\,\Sigma_f(\cdot)\Big),\;\;\mu_f(\cdot) := f(\cdot,\mu_w),\;\;\Sigma_f(\cdot) := J^f_{\mu_w}(\cdot)\Sigma_w J^f_{\mu_w}(\cdot)^T.\;\;(5)$$

This linear approximation of the functional output Eqs. (4) and (5) leads to, when further approximated with a finite context set $X_I$, the following closed-form expression for the KL divergence term:

$$\mathrm{KL}\big(q(f(X_I,\Theta))\|p(f(X_I,\Theta))\big) \approx \mathrm{KL}\big(\mathcal{N}(\mu_{f_q}(X_I),\Sigma_{f_q}(X_I))\|\mathcal{N}(\mu_{f_p}(X_I),\Sigma_{f_p}(X_I))\big), \quad (6)$$

where $(\mu_{f_q}(X_I),\Sigma_{f_q}(X_I))$ are the mean and covariances of the variational functions evaluated at $X_I$ and $(\mu_{f_q}(X_I),\Sigma_{f_q}(X_I))$ are defined similarly for prior functions. It is common to further restrict the covariances to be diagonal, allowing the joint KL divergences to decompose into a sum of instance-wise KL divergences.

## 3 DIFFICULTY OF FUNCTION-SPACE PRIOR SPECIFICATION

While the function space prior induced by the weight space prior results in a tractable computation, it comes with a cost of less interpretability in the prior. Especially, unlike the regression task where Gaussian process (GP) can be used to build the interpretable function space prior by specifying the kernel function (Flam-Shepherd et al., 2017; Karaletsos & Bui, 2020; Tran et al., 2022), it has been less clear to specify the interpretable function space prior for the classification task.

A default function space prior would be a uniform prior by setting the zero-mean Gaussian prior on function space, i.e., $p(f(\cdot)) = N(f(\cdot); \mathbf{0}, \sigma^2 I)$ with all-zero vector $\mathbf{0} = [0,..,0] \in R^Q$ for any input $\cdot \in \mathcal{X}$. This input-independent function space prior induces that the predictive probability computed from a BNN, are regularized to stay somewhat close to a uniform distribution due to $\mathrm{softmax}(\mathbf{0}) = [1/Q,...,1/Q] \in \Delta^{Q-1}$. However, the corresponding regularization could be suboptimal, because it

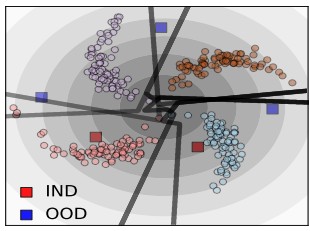
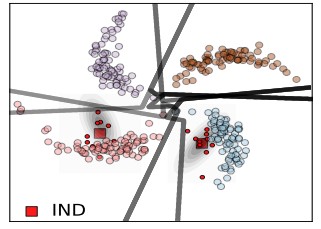
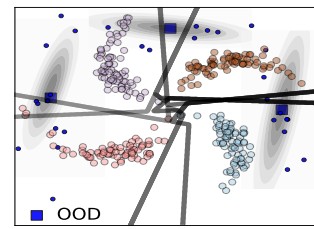

(a) Uniform function space prior    (b) IND function space prior    (c) OOD function space prior

Figure 1: Example of Function Space prior with 4-class pinwheel classification: In each panel, circle points denote training sets, and two black lines denote 2 decision boundaries of the linear classifiers. Three blue square points (■) and two red square points (■) denote auxiliary inputs, which are far from training points and are placed closely with the training sets, respectively. **Panel (a)** depicts the uniform function space prior by setting the function space prior as $p(f(\cdot)) = \mathcal{N}(f(\cdot); \mathbf{0}, \sigma^2 I)$, which induces all context inputs to be uniformly classified into any class. **Panel (b)** and **(c)** depict the informative function space prior, motivated by BMA; each predictive sample function on IND set (●) is assigned to be specific class, whereas each prediction on OOD set (●) has more disagreements.

encourages the predictions of the data from in-distribution (IND) set unnecessarily uncertain while allowing the predictions of the data from out-of-distribution (OOD) set uncertain.

One may elicit a weight-space induced prior as in Rudner et al. (2022), where the the function space prior $p(f(\cdot, \Theta))$ is induced by specifying the parameters $(\mu_w, \Sigma_w)$ in weight-space prior $\mathcal{N}(\Theta; \mu_w, \Sigma_w)$ in Eq. (5). As mentioned earlier, while this may provide a tractable inference algorithm, the interpretation of the derived function-space prior is rather obscure. Specifically, when $\mathcal{N}(\Theta; \mu_w, \Sigma_w)$ is set to be a zero-mean Gaussian distribution, i.e., $\mu_w = \mathbf{0}$ and $\Sigma_w = I$, this results in the function space prior with the zero-mean but input-dependent variance, i.e., $p(f_{\text{lin}}(\cdot)) = \mathcal{N}(f_{\text{lin}}(\cdot); \mathbf{0}, \Sigma_{f_p}(\cdot))$ by considering $\mu_{f_p}(\cdot) = f(\cdot, 0) = \mathbf{0}$ and $\Sigma_f(\cdot) = J^f_{\mu_w}(\cdot)J^f_{\mu_w}(\cdot)^T$ in Eq. (5). This prior raise a question of how uncertain the variance on the function space can make the predictions of the data from in and out-of-distribution dataset. For other function space prior induced from the non-zero mean weight-space Gaussian prior $\mathcal{N}(\Theta; \mu_w, \Sigma_w)$, we are not sure by what mechanism the prior may encourage the predictions of the data from both in and out-of-training set.

## 4    REFINEMENT OF FUNCTION-SPACE VARIATIONAL INFERENCE

In this section, we focus on building the informative function space prior for the classification task, and present a refined function-space variational inference[2] to alleviate the issue described above.

**Rethinking the role of function-space prior from the standpoint of model averaging.** To establish what is a good function space prior for the purpose of classification, let us reconsider the role of the function space posterior $p(f(\cdot) \mid \mathcal{D})$ computed from the prior distribution $p(f(\cdot))$. When a test input $x_*$ is given, we consider the predictive distribution via Bayesian Model Averaging (BMA):

$$p(y_* \mid x_*, \mathcal{D}) = \int p(y_* \mid f(x_*))p(f(x_*) \mid \mathcal{D})df(x_*) \approx \frac{1}{J}\sum_{j=1}^{J} p(y_* \mid f_{(j)}(x_*)), \quad (7)$$

where $\{f_{(j)}(\cdot)\}_{j=1}^{J} \overset{\text{i.i.d.}}{\sim} p(f(\cdot) \mid \mathcal{D})$ denotes $J$ sample functions of posterior distribution, and each $f_{(j)}(x_*) = [f_{(j)}(x_*)_1, \ldots, f_{(j)}(x_*)_Q] \in \Delta^{Q-1}$ represents the $Q$-dimensional predictive probability to classify the test input $x_*$. Ideally, when $x_*$ is close to training data $\mathcal{D}$, we expect the corresponding predictive distribution $p(y_* \mid x_*, \mathcal{D})$ to have high peak value $f_{(j)}(x_*)_q$ for some specific class $q$ (smaller entropy). On the other hand, if $x_*$ is more of an OOD data, we expect $p(y_* \mid x_*, \mathcal{D})$ to be closer to the uniform distribution (higher entropy). The existing uniform function-space prior achieves this by making each predictive probability $f_{(j)}(x_*)$ to have higher entropy, not by directly

---

[2]Although we do not cover the case of the regression task in this work, our approach can be extended for the regression task as well if binning the real-valued function space, i.e., $\mathcal{Y} \subset \sum_q b_q$ with the finite $Q$ bins $\{b_1, \ldots, b_Q\}$, and regarding the feature of the penultimate layer $h(\cdot)$ to have the index of the bin as the label.

increasing the entropy of *BMA prediction*. As a side effect, this regularization would make the prediction to have high entropy (thus less certain) for a test input $x_*$ sufficiently close to training set.

Instead, we may design a function space prior that does not explicitly encourage generating high-entropy predictions for each predictive probability $f_{(j)}(x_*)$, but the averaged prediction (via BMA) would still have high entropy when encountered with an OOD input $x_*$. A plausible situation is when the individual predictions $f_{(j)}(x_*)$ may have highly peaked distributions even for an OOD input $x_*$, but the predicted classes for different samples *disagree with each other*, giving a high-entropy prediction when averaged via BMA. In other words, what we expected from our function space prior is not to induce the individual predictive distribution $f_{(j)}(x_*)$ to be uniform distribution, but to be peaked distribution as well as to allow their varying *degree of disagreements* depending on how the $x_*$ deviates from the training set $\mathcal{D}$. We argue that this way of constructing prior can provide a better trade-off between making the BNNs certain for IND-like data and yet being uncertain for OOD-like data. We provide a conceptual description of our argument; Fig. 1a depicts the uniform function-space prior, and Figs. 1b and 1c depict the refined function-space prior.

**Refined function-space prior distribution.**    To build the function space prior following the intuition described above, we adapt the empirical Bayes that learns the parameters of the prior distribution from the training set (Casella, 1992). We assume the first $L-1$ layers $\{f^l\}_{l=1}^{L-1}$ to be deterministic layers and the last $L$-th layer $f^L$ to be Bayesian MLP layer for easy Jacobian computation without the use of automatic differentiation (Horace He, 2021); see the details in Appendix A.2. We first train the parameters of $\{f^l\}_{l=1}^{L}$ during the initial training stages, and employ the hidden feature $h(\cdot) = (f^{L-1} \circ \cdots \circ f^2 \circ f^1)(\cdot) \in R^h$ and the weight parameters $\Theta^L \in R^{Q \times h}$ for the last-layer $f^L$, obtained at the pre-determined iterations in this period.

Specifically, at the pre-determined iterations $t \in \mathcal{T} := \{t_1, .., t_T\}$, we compute the empirical mean $\hat{m}_h^q \in R^h$ for $q$-th label and the shared covariance $\hat{S}_h \in R^{h \times h}$ defined as:

$$\hat{m}_h^q = \frac{1}{N^q} \sum_{\{i; y_i = q\}} h(x_i), \;\; \hat{S}_h = \frac{1}{N} \sum_q \sum_{\{i; y_i = q\}} \Delta_q h(x_i) \, \Delta_q h(x_i)^T, \;\; \Delta_q h(\cdot) := h(\cdot) - \hat{m}_h^q \quad (8)$$

From iteration $t_T + 1$, we compute the aggregated empirical mean of hidden feature $\hat{m}_h^q$, its shared covariance $\hat{S}_h$, the empirical mean of last-layer weight $\hat{\mu}_w^L$, and its covariance $\hat{\Sigma}_w^L$, defined as:

$$\text{feature level (h)}: \;\; \hat{m}_h^q = \frac{1}{T} \sum_{t=1}^{T} \hat{m}_h^q(t), \qquad \hat{S}_h = \frac{1}{T} \sum_{t=1}^{T} \hat{S}_h(t),$$

$$\text{weight level (w)}: \;\; \hat{\mu}_w^L = \frac{1}{T} \sum_t \Theta^L(t), \qquad \hat{\Sigma}_w^L = \frac{1}{T} \sum_t \left( \Theta^L(t)^2 - (\hat{\mu}^L)^2 \right), \quad (9)$$

where the empirical parameter $\hat{m}_h^q(t)$ and $\hat{S}_h(t)$, defined in Eq. (8), and the last layer weight parameter $\Theta^L(t)$ are obtained at $t$-th iteration for $t \in \mathcal{T}$. Then, with the random weight parameters $\Theta^L \sim \mathcal{N}(\Theta^L \mid \hat{\mu}_w^L, \text{diag}(\hat{\Sigma}_w^L))$, we define the random linear function $f_{\text{BMA}}(\cdot)$ for a input $\cdot \in \mathcal{X}$:

$$f_{\text{BMA}}(\cdot) = \Theta^L \, \text{proj}(h(\cdot)), \quad \text{proj}(h(\cdot)) = \sum_q \left( \frac{\exp\left(-\|\Delta_q h(\cdot)\|_{\hat{S}_h^{-1}}^2\right)}{\sum_q \exp\left(-\|\Delta_q h(\cdot)\|_{\hat{S}_h^{-1}}^2\right)} \right) \hat{m}_h^q, \quad (10)$$

where $\|\Delta_q h(\cdot)\|_{\hat{S}_h^{-1}}^2 := \Delta_q h(\cdot)^T \text{diag}(\hat{S}_h^{-1}) \Delta_q h(\cdot)$ denotes the distance of $h(\cdot)$ from the $q$-th component empirical parameters of hidden feature $(\hat{m}_h^q, \hat{S}_h)$ in Eq. (9). Roughly speaking, $f_{\text{BMA}}(\cdot)$ in Eq. (10) first projects $h(\cdot)$ as the representation of the empirical hidden feature $\text{proj}(h(\cdot))$. Then, by using the linear mapping with the parameters $(\hat{\mu}_w^L, \hat{\Sigma}_w^L)$ in Eq. (9), we build the function space prior $p(f_{\text{BMA}}(\cdot)) = \mathcal{N}(f_{\text{BMA}}(\cdot); \mu_{f_p}(\cdot), \Sigma_{f_p}(\cdot))$ with following parameters:

$$\mu_{f_p}(\cdot) = \hat{\mu}_w^L \, \text{proj}(h(\cdot)), \quad \Sigma_{f_p}(\cdot) = \text{diag}\left( \left[ \|\text{proj}(h(\cdot))\|_{\hat{\sigma}_1^2}^2, \, ... \, , \|\text{proj}(h(\cdot))\|_{\hat{\sigma}_Q^2}^2 \right] \right), \quad (11)$$

where $\hat{\sigma}_q^2 \in R^h$ denotes the $q$-th row of the variance parameters $\hat{\Sigma}_w^L := [\hat{\sigma}_1^2, .., \hat{\sigma}_Q^2] \in R^{Q \times h}$. This function space prior follows the concept described in Figs. 1b and 1c because when the context input $\cdot \in X_{\mathcal{I}}$ is close to IND-like set, the $\text{proj}(h(\cdot))$ would be projected to one element out of $\{\hat{m}_h^q\}_{q=1}^{Q}$ and then allows the $\mu_{f_p}(\cdot)$ to be the high peaked logits (low entropy); see the details in Appendix A.3.

**Context inputs from adversarial hidden features.** Without introducing an additional datasets for the context inputs, we propose to build the hidden feature of context inputs $h(\cdot)$ by directly using the training set $\mathcal{D}$. For a pair of the training set $(x_i, y_i) \in \mathcal{D}$, we consider $h(\cdot)$ to be the adversarial hidden feature using the distance $\|\Delta_q h\|^2_{\hat{S}_h^{-1}}$, defined in Eq. (10), as follows:

$$h(\cdot) \coloneqq \underset{h \in B_r(h(x_i))}{\arg\max} \left( \min_q \|\Delta_q h\|^2_{\hat{S}_h^{-1}} \right), \qquad (12)$$

where $B_r(h(x_i)) \coloneqq \{h; \|h - h(x_i)\|_2 \le r\}$ denotes the $r$-neighborhood of the $h(x_i)$. The context hidden feature $h(\cdot)$ in Eq. (12) is designed to incorporate the function space prior $p(f_{\mathrm{BMA}}(\cdot))$ on the boundary of the training hidden features $\{h(x_i); (x_i, y_i) \in \mathcal{D}\}$ using the the Mahalanobis distance $\min_q \|\Delta_q h\|^2_{\hat{S}_h^{-1}}$ representing the shortest distance of $h(\cdot)$ over $Q$ classes. This context hidden feature $h(\cdot)$ is used to compute $p(f_{\mathrm{BMA}}(\cdot))$ for KL regularization by putting $h(\cdot)$ into $\mathrm{proj}(h(\cdot))$ in Eq. (10).

**Refined function-space variational distribution.** We refine the variational distribution $q(f(\cdot))$ to allow the sample functions to be peaked distribution as well as have the varying degree of disagreements depending on how the predictive input $\cdot \in \mathcal{X}$ deviates from the training set.

To this end, we introduce the categorical latent variable $z(\cdot) = (z_1, .., z_Q) \in \{0, 1\}^Q$, with $\sum_Q z_Q = 1$, that represents how uncertain the feature $h(\cdot)$ is used according to the empirical distribution of $h(\cdot)$, to build the sample function $f(\cdot)$ having the different degree of disagreement for a input $\cdot \in \mathcal{X}$. For the variational distribution of the last layer weight parameter $q(\Theta^L) = \mathcal{N}(\Theta^L; \mu_{w_q}^L, \Sigma_{w_q}^L)$, we define $q(z(\cdot)|h(\cdot)) = \mathrm{Cat}\left(z(\cdot) \mid \hat{p}(\cdot)\right)$ with the parameter $\hat{p}(\cdot)$, defined as:

$$\hat{p}(\cdot) = \mathrm{Softmax}\left( \mu_f(\cdot) \,/\, \sqrt{1 + \pi/8 \,\|\Delta h(\cdot)\|^2_{\hat{S}_h^{-1}}}, \right) \in \Delta^{Q-1}, \qquad (13)$$

where $\mu_f(\cdot) \coloneqq \mu_{q_w}^L h(\cdot) \in R^Q$ and $\|\Delta h(\cdot)\|^2_{\hat{S}_h^{-1}} \coloneqq \left[\|\Delta_1 h(\cdot)\|^2_{\hat{S}_h^{-1}}, .., \|\Delta_Q h(\cdot)\|^2_{\hat{S}_h^{-1}}\right] \in R_+^Q$ with $\Delta_q h(\cdot)$ in Eq. (8). Based on multi-dimensional probit approximation (MPA) (Gibbs, 1998; Lu et al., 2020), the $\hat{p}(\cdot)$ is obtained by approximately marginalizing the random function $\tilde{f}$ over the following Gaussian distribution, parameterized by the empirical distance of $h(\cdot)$ as follows:

$$\hat{p}(\cdot) \approx \int \mathrm{Softmax}(\tilde{f}(\cdot)) \, \mathcal{N}\left(\tilde{f}(\cdot) \,;\, \mu_f(\cdot), \mathrm{diag}\left(\|\Delta h(\cdot)\|^2_{\hat{S}_h^{-1}}\right)\right) d\tilde{f}(\cdot). \qquad (14)$$

This $\hat{p}(\cdot)$ controls the degree of the disagreement of the predicted classes by encoding what predictive class is more likely to be in the nominator of $\hat{p}(\cdot)$ and how far $h(\cdot)$ deviates from the empirical distribution of $h(\cdot)$ in the denominator of $\hat{p}(\cdot)$. When modeling the high dimensional output, the latent variable $z(\cdot)$ can be sampled from the marginalized top-$k$ distribution of $z(\cdot)$ by choosing $k$ indexes of $\hat{p}(\cdot)$ having the top $k$ largest values of $\hat{p}(\cdot)$, and marginalizing the left redundant $C - k$ dimensions.

Next, with temperature parameter $\tau \in (0, 1)$ and all-one vector $\mathbf{1} = [1, .., 1] \in R^Q$, we define the conditional variational functional distribution $q\left(f_{\mathrm{lin}}(\cdot) \mid z(\cdot), h(\cdot), \tau\right) = \mathcal{N}\left(f_{\mathrm{lin}}(\cdot) \,;\, \mu_{f|z}(\cdot), \Sigma_{f|z}(\cdot)\right)$:

$$\mu_{f|z}(\cdot) = (\mathbf{1} + \tau z(\cdot)) \, \mu_{f_q}(\cdot), \qquad \Sigma_{f|z}(\cdot) = \mathrm{diag}(\mathbf{1} + \tau z(\cdot)) \circ \Sigma_{f_q}(\cdot), \qquad (15)$$

that are intended to assign the peaked logits value on the randomly chosen dimension of the function output via latent variable $z(\cdot)$. This latent modeling constructs $\{f_{(j)}(\cdot)\}_{j=1}^J$ by first sampling the latent variable $\{z_{(j)}(\cdot)\}_{j=1}^J$ from the peaked or flat categorical distribution according to the denominator $\|\Delta h(\cdot)\|^2_{\hat{S}_h^{-1}}$ in Eq. (13), and then sampling $f_{(j)}(\cdot)$ with $\epsilon_j \sim \mathcal{N}(\epsilon; 0, I) \in R^Q$ over function space:

$$z_{(j)}(\cdot) \sim q(z(\cdot)|h(\cdot)), \quad f_{(j)}(\cdot) \sim q(f_{\mathrm{lin}}(\cdot)|z_{(j)}(\cdot), h(\cdot), \tau), \quad \text{with } f_{(j)}(\cdot) = \mu_{f|z_{(j)}}(\cdot) + \Sigma_{f|z_{(j)}}^{1/2}(\cdot) \, \epsilon_j$$

## 5 EXPERIMENTS

In this section, we study the following questions through empirical validation: **(1)** Does the use of the uniform function space prior really lead to a trade-off in performance between the IND and OOD datasets? - Section 5.1, **(2)** Are the proposed refinements, motivated by BMA, effective in improving uncertainty estimation for the both IND and OOD datasets? - Section 5.2, and **(3)** Does the proposed inference yield reliable uncertainty estimates for covariate shift and downstream datasets? - Sections 5.3 and 5.4

**Experiment Setting.** We basically use the widely-used deep learning architecture, such as ResNet (He et al., 2016), as our base model. Then, we convert the model into a last-layer BNNs by replacing the last MLP layer with a Bayesian MLP layer. This approach is motivated by the proven effectiveness of a partial Bayesian approach for Deep Neural Networks (DNN), as compared to a fully Bayesian approach applied to all layers of the DNN (Kristiadi et al., 2020; Sharma et al., 2023).

For evaluating the trained model, we measure the test accuracy (ACC), calibrated negative log likelihood (NLL), and calibrated expected calibration error (ECE) on the IND test set as indicators of uncertainty estimation performance for the IND set. To this end, we first find the calibrated temperature for NLL and apply this temperature to each metric, following the guidance for uncertainty estimation evaluation in (Ashukha et al., 2020). Also, we measure the Area Under the Receiver Operating Characteristic (AUROC) on the OOD set, serving as indicators of performance for OOD set. We use the calibrated predictive entropy as the input and the IND set's status as the label.

## 5.1 TRADE-OFF ISSUE OF THE UNIFORM FUNCTION SPACE PRIOR REGULARIZATION

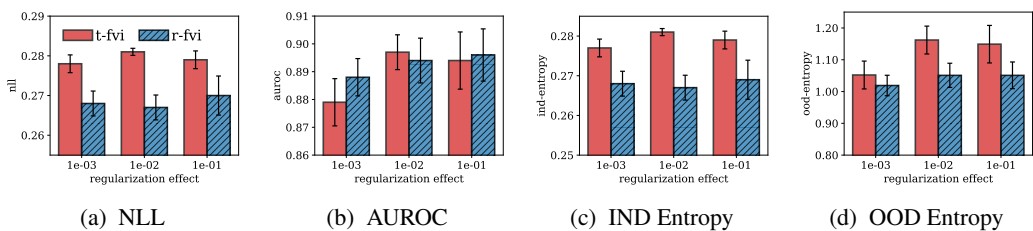

|  (a)  NLL  |  (b)  AUROC  |  (c)  IND Entropy  |  (d)  OOD Entropy |

Figure 2: Investigation on uniform function space prior: we run 5 experiments with different random seeds and report the mean and one-standard error for each metric. **Panel (a)** shows the calibrated NLL on CIFAR 10 over varying hyperparameter $\lambda$ for KL term. **Panel (b)** shows the AUROC on SVHN set. **Panel (c)** and **Panel (d)** show the predictive entropy on CIFAR 10 and SVHN, respectively.

We set CIFAR 10 as training set and CIFAR 100 as context inputs, and train the model by using the ELBO $\mathcal{L}(\Theta_f)$ in Eq. (2) along with the uniform function space prior as $p(f_{\text{lin}}(\cdot)) = \mathcal{N}(f_{\text{lin}}(\cdot); 0, 10I)$. We consider the varying hyperparameters $\lambda \in \{10^{-3}, 10^{-2}, 10^{-1}\}$ in Eq. (3) to investigate the effect of the uniform function space prior via the regularization on the uncertainty estimation for IND and OOD set. We control $\lambda$ as the relative ratio between likelihood and KL divergence to apply the same amount of the regularization to the model, regardless of the scale of KL term; $\lambda = 10^{-1}$ means that the value of KL term in Eq. (5) is adaptively rescaled to be $1/10$ of the likelihood over iterations.

**Results.** Figs. 2a and 2b describe the calibrated NLL on CIFAR 10 (IND) and AUROC on SVHN (OOD), respectively. Figs. 2c and 2d describe the predictive entropy for CIFAR 10 and SVHN, respectively, used as the input feature for calculating the AUROC. Fig. 2 implies that when applying the stronger regularization of uniform function space prior (T-FVI) to the model during training, the NLL deteriorates, but AUROC improves as shown in Figs. 2a and 2b. Also, its predictive entropy exhibits higher values than that of the proposed inference (R-FVI), and increases for both IND and OOD sets as shown Figs. 2c and 2d. On the other hand, the NLL of the R-FVI increases slightly and the corresponding AUROC becomes as accurate as that of the baseline. Notably, the predictive entropy for IND set exhibits small variations compared to that of T-FVI. These results suggest that using the uniform function space prior may potentially lead to a trade-off in uncertainty estimation between IND and OOD sets, and the proposed inference method can mitigate this issue.

## 5.2 BENCHMARK EXPERIMENT

As following the experimental setup conducted in (Rudner et al., 2022), we use CIFAR 10 and CIFAR 100 as the IND set and SVHN as the OOD set. We compare the proposed inference with other baseline inference methods: Maximum a posterior (MAP), DUQ (Van Amersfoort et al., 2020), SWAG (Maddox et al., 2019), MFVI (Blundell et al., 2015), and T-FVI (Rudner et al., 2022) using a cold posterior effect. For variational inference methods, we employ the MAP inference for the first 80 percentage of training iterations, and apply the variational inference for remaining iterations; further details can be found in Appendix B.3.

| Model / Data | Method | Context set | ACC ↑ | NLL ↓ | ECE ↓ | AUROC ↑ |
|---|---|---|---|---|---|---|
| ResNet 18 CIFAR 10 | MAP | | (**0.951**, 0.001) | (**0.165**, 0.002) | (0.012, 0.001) | (0.948, 0.005) |
| | DUQ* | | (0.941, 0.002) | — | — | (0.927, 0.013) |
| | MFVI | | (0.949, 0.001) | (0.189, 0.003) | (0.011, 0.001) | (0.942, 0.009) |
| | SWAG* | | (0.931, 0.001) | — | (0.067, 0.002) | (0.898, 0.005) |
| | T-FVI | CIFAR 100 | (0.949, 0.002) | (0.176, 0.008) | (0.012, 0.001) | (0.938, 0.012) |
| | **R-FVI** | CIFAR 100 | (0.950, 0.001) | (0.169, 0.003) | (0.011, 0.000) | (0.955, 0.003) |
| | | ADV-h | (**0.951**, 0.000) | (0.167, 0.001) | (**0.010**, 0.001) | (**0.957**, 0.002) |
| ResNet 18 CIFAR 100 | MAP | | (0.772, 0.000) | (0.932, 0.002) | (0.051, 0.002) | (0.865, 0.025) |
| | MFVI | | (0.770, 0.002) | (1.035, 0.010) | (0.049, 0.003) | (0.862, 0.019) |
| | T-FVI | T-ImageNet | (**0.779**, 0.003) | (0.914, 0.002) | (0.049, 0.002) | (0.874, 0.013) |
| | **R-FVI** | T-ImageNet | (0.775, 0.003) | (0.912, 0.011) | (**0.044**, 0.000) | (**0.890**, 0.030) |
| | | ADV-h | (0.778, 0.001) | (**0.891**, 0.003) | (0.047, 0.004) | (0.883, 0.014) |

Table 1: Benchmark experiment results: we report the mean and one-standard deviation of each metric over different 3 random seeds. The results of Bayesian inference are obtained by using 100 sample functions, i.e., $\{f_{(j)}(x_*)\}_{j=1}^{100}$ in Eq. (7).The results of * are taken from (Rudner et al., 2022).

**Results.** Table 1 shows that our approach of R-FVI, achieves better NLL and AUROC compared to T-FVI and MAP. This implies that the proposed inference performs better in terms of uncertainty estimation for both the IND and OOD sets. Also, using the adversarial context hidden feature (ADV-h) without using the additional context inputs can leads to comparable uncertainty estimation.

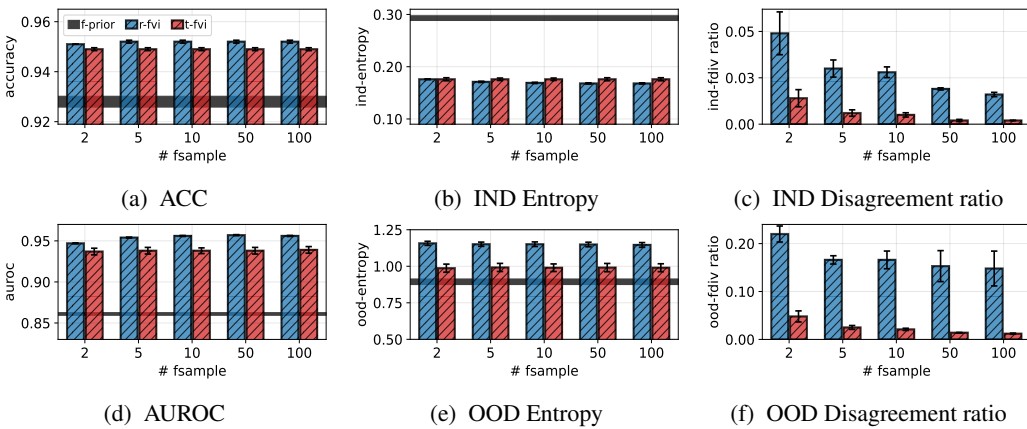

(a) ACC  (b) IND Entropy  (c) IND Disagreement ratio

(d) AUROC  (e) OOD Entropy  (f) OOD Disagreement ratio

Figure 3: Effect of the proposed inference over varying $J$ sample functions: for the results of proposed prior $f_{\mathrm{BMA}}(\cdot)$, we use the predictive probability as $\mathrm{Softmax}(\mu_{f_p}(\cdot)/\sqrt{1+\pi/8\Sigma_{f_p}(\cdot)})$ with its parameters in Eq. (11). We depict the confidence interval with one-standard error.

Fig. 3 investigates the effect of the proposed tricks by using the trained ResNet 18 on CIFAR 10. We describe the results of the function space prior $f_{\mathrm{BMA}}(\cdot)$ evaluated on both the IND (CIFAR 10) and OOD (SVHN) set. We also compare the degree of the sample function disagreement via the disagreement ratio $(J - B)/B$, where $B$ denotes the number of the most dominant predictive class out of $J$ sample functions. The proposed prior adaptively induces the informative prior depending the status of dataset; note its varying predictive entropy in Figs. 3b and 3e and the subsequent results in Figs. 3a and 3d. Furthermore, as the number of sample function increases, the performances of R-FVI improve significantly, as shown in Figs. 3a and 3d. Notably, its disagreement ratio exhibits a significant distinction between the IND and OOD sets, whereas that of T-FVI approaches zero in both sets, as demonstrated in Figs. 3c and 3f.

Fig. 4 compares the example of the predictive probabilities for IND and OOD set, confirming that the predictions of R-FVI have the significant disagreement on OOD set compared to those of the T-FVI.

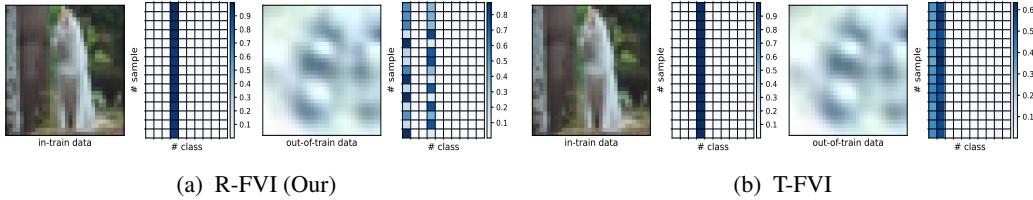

(a) R-FVI (Our)  (b) T-FVI

Figure 4: Comparison of 15 sampled predictive probabilities for IND (CIFAR 10) and OOD (SVHN).

## 5.3 UNCERTAINTY ESTIMATION ON COVARIATE SHIFT DATASET.

We validate the proposed inference by evaluating the uncertainty estimation performance on the corrupted CIFAR 10 (Hendrycks & Dietterich, 2018), which has been widely used for evaluation on covariate shift. Due to limited space, we attach our results in Appendix B.4 showing the proposed inference results in reliable uncertainty estimation on corrupted datasets as well.

## 5.4 TRANSFER LEARNING WITH VISION TRANSFORMER.

We validate that the proposed inference can transfer the large-sized pre-trained model to adapt to downstream task. We use the pre-trained VIT-Base model (16 patch and 224 resolution) on the ImageNet 21K [3], and consider the last-layer BNN as done in ResNet. We use $J = 100$ for the BMA.

**Results.** Table 2 shows that proposed inference can lead to reliable uncertainty estimation on IND and OOD set as adapting the large-sized VIT model (#parameters $= 86.6M$) to downstream task.

| Method | Context set | ACC ↑ | NLL ↓ | ECE ↓ | AUROC ↑ |
|--------|-------------|-------|-------|-------|---------|
| MAP | | (0.909, 0.001) | (0.333, 0.004) | (0.018, 0.001) | (0.961, 0.004) |
| MFVI | | (0.908, 0.002) | (0.325, 0.014) | (0.015, 0.003) | (0.970, 0.007) |
| T-FVI | T-ImageNet | (0.908, 0.001) | (0.328, 0.004) | (0.014, 0.001) | (0.952, 0.002) |
| **R-FVI** | T-ImageNet | (**0.914**, 0.001) | (**0.313**, 0.001) | (0.016, 0.001) | (**0.970**, 0.004) |
| | ADV-h | (0.913, 0.001) | (0.315, 0.001) | (**0.014**, 0.001) | (0.967, 0.004) |

Table 2: Transfer learning with VIT; these results from 3 experiments are reported.

## 6 RELATED WORK & CONCLUSION

**Function space prior:** As our variation inference method employs the empirical parameters of hidden features and the last-layer weight parameters, using the well-known inductive bias of the DNN (Lee et al., 2018; Maddox et al., 2019), to set the parameters of the prior distribution, our approach can be regarded as a empirical Bayes method. This is analogous to the approach taken by Immer et al. (2021) that uses empirical Bayes for the model selection of the BNNs. However, our method distinguishes itself from (Immer et al., 2021) by establishing the prior in function space rather than weight space. Also, our approach differs from the existing function space prior approaches that predefine priors, such as the Gaussian process prior (Flam-Shepherd et al., 2017; Tran et al., 2022), noise constrictive priors (Hafner et al., 2020), and physics informed priors (Meng et al., 2022).

**Variational Inference:** Our work aim at improving the prediction performance of BNNs based on the principle of BMA. Thus, we propose the variational function space distribution to sample useful predictive functions in sense of BMA, whereas the existing works mainly have focused on addressing tractability issue for the functional KL divergence (Matthews et al., 2016; Burt et al., 2020) and presented tractable inference algorithms (Sun et al., 2019; Rodrguez-Santana et al., 2022; Rudner et al., 2022) to handle the issue.

**Conclusion.** In this work, we note that the existing function space BNNs have an difficulty in incorporating the function space prior.To tackle this issue, we propose the informative function space prior and the variational distribution on function space by emploting the observation from the BMA.

---
[3]https://github.com/huggingface/pytorch-image-models

**Ethics statement.** This paper does not include any ethical issues. This work introduces the way to improve the function space variational inference, which has nothing to do with the ethics issue.

**Reproducibility statement.** We describe our algorithm and the hyperparameter used for the experiments in Appendices A.1 and B.1.

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

# A APPENDIX: METHODOLOGY DETAILS

## A.1 ALGORITHM

We describe the algorithm of the proposed inference in Algorithm 1. Our implementation is available at https://anonymous.4open.science/r/RefinedFSBayes-B8F5.

---

**Algorithm 1** Improved Function-Space Variational Inference with Informative Priors

---

**Require:** training iterations $T$, pre-determined iterations $\mathcal{T} = \{t_1, .., t_T\}$
first $L - 1$ layers parameters $\{\Theta^{(l)}\}_{l=1}^{L-1}$, and the last $L$-th Bayesian layer parameters $(\mu_w, \Sigma_w)$
  **for** $t = 1, \ldots, T$ **do**
    **if** $t \leq t_T$ **then**                                           ▷ Training by MAP Inference
      Train $\{\Theta^{(l)}\}_{l=1}^{L-1}$ and $\mu_w$ with function space ELBO $\mathcal{L}_{\text{fvi}}$ of Eq. (2) without regularization
      **if** $t \in \mathcal{T}$ **then**                                       ▷ Preparation for $f_{\text{BMA}}(\cdot)$
        Compute the empirical parameters of $(\hat{m}_h^q, \hat{S}_h)$ and $(\hat{\mu}_w^L, \hat{\Sigma}_w^L)$ in Eq. (9)
      **end if**
    **else**                                 ▷ Training by Function Space Variational Inference
      Construct the adversarial hidden feature $h(\cdot)$ in Eq. (12)
      Evaluate the BMA function prior distribution $p(f_{\text{BMA}}(\cdot))$ in Eq. (10)
      Build the sample functions $\{f_{(j)}(\cdot)\}_{j=1}^J$ by applying RP trick on funtion sapce with Eq. (15)
      Train the parameters $\{\Theta^{(l)}\}_{l=1}^{L-1}$ and $(\mu_w, \Sigma_w)$ with the function space ELBO $\mathcal{L}_{\text{fvi}}$ of Eqs. (2) and (5).
    **end if**
  **end for**

---

## A.2 COMPUTATION OF FUNCTION SPACE DISTRIBUTION WITHOUT JACOBIAN COMPUTATION

Let $f(\cdot, \Theta) \in R^Q$ be a $L$-layers BNN with the random weight parameters $\Theta = \{\Theta^{(l)}\}_{l=1}^L$, as follows:

$$f(\cdot, \Theta) = \left(f^{(L)} \circ \cdots \circ f^{(2)} \circ f^{(1)}\right)(\cdot), \quad \text{and} \quad f^{(l)}(\cdot) = \sigma(\Theta^{(l)} [\cdot; 1]), \tag{16}$$

where $\Theta^{(l)}$ denotes $l$-th layer random weight parameters, and $\sigma(\cdot)$ denotes the activation function. For the brevity, we omit the bias term of each $\Theta^{(l)}$, which does not break the logic of our statement. Additionally, we assume the structure of the $f(\cdot, \Theta)$ to compute parameters of function space distribution without the Jacobian computation via the automatic differentiation (Horace He, 2021), which has known to incur the out-of memory issue, as follows:

- The first $L - 1$ layers $\{f^l\}_{l=1}^{L-1}$ are assumed to be deterministic layers with the weight parameters $\{\Theta^{(l)}\}_{l=1}^{L-1}$, understood as $p(\Theta^{(l)}) = \delta_{\Theta^{(l)}}(\Theta')$.

- The last $L$-th layer $f^L$ is assumed to be a Bayesian MLP layer with the random parameter $\Theta^{(L)}$ following $p(\Theta^{(L)}) = \mathcal{N}(\Theta^{(L)}; \mu_w, \Sigma_w)$. We notate each parameter $\Theta^{(L)} := [\theta_1, .., \theta_Q] \in R^{Q \times h}$, $\mu_w := [\mu_1, .., \mu_Q] \in R^{Q \times h}$ and $\Sigma_w := [\sigma_1^2, .., \sigma_Q^2] \in R_+^{Q \times h}$, where $\theta_q, \mu_q \in R^h$, and $\sigma_q^2 \in R_+^h$ denote the $q$-th row vector of each matrix, respectively.

Then, the $f(\cdot, \Theta)$ in Eq. (16) can be understood as follows:

$$f(\cdot, \Theta) = \Theta^{(L)} h(\cdot) = [\theta_1^T h(\cdot), .., \theta_Q^T h(\cdot)], \quad \text{with} \quad h(\cdot) := (f^{L-1} \circ \cdots \circ f^2 \circ f^1)(\cdot) \in R^h, \tag{17}$$

where $h(\cdot)$ has a deterministic values as the output.

For the computation of the Jacobin matrix $J_{\mu_w}^f(\cdot) := [\frac{\partial f}{\partial \Theta^{(L)}}]_{\Theta^{(L)} = \mu_w} \in \mathbb{R}^{Q \times P}$, with the number of the parameter $P = Q \times h$, we can compute the $q$-th row of Jacobian matrix as $\left[J_{\mu_w}^f(\cdot)\right]_q \in R^P$:

$$\left[J_{\mu_w}^f(\cdot)\right]_q = \left[\frac{\partial(\theta_q^T h(\cdot))}{\partial \theta_1}, ...., \frac{\partial(\theta_q^T h(\cdot))}{\partial \theta_q}, ...., \frac{\partial(\theta_q^T h(\cdot))}{\partial \theta_Q}\right] = \left[\underbrace{\mathbf{0}}_{\text{1-th}}, ..., \underbrace{h(\cdot)}_{q\text{-th}}, ..., \underbrace{\mathbf{0}}_{Q\text{-th}}\right],$$

$$\tag{18}$$

that has the non-zero entries as $h(\cdot)$ in $q$-th block and zero entries $\mathbf{0} \in R^h$ in left blocks. This allows one to compute the covariance $\Sigma_f(\cdot)$ of the functions space distribution in Eq. (5), as follows:

$$[\, \Sigma_f(\cdot) \,]_{qp} = [\, J^f_{\mu_w}(\cdot) \, \mathrm{diag}(\, \mathrm{vec}(\Sigma_w) \,) \, J^f_{\mu_w}(\cdot)^T \,]_{qp} \tag{19}$$

$$= \underbrace{h(\cdot)^T \, \mathrm{diag}(\sigma^2_q) \, h(\cdot)}_{:=\|h(\cdot)\|^2_{\sigma^2_q}} \, \mathbf{1}_{q=p} = \|h(\cdot)\|^2_{\sigma^2_q} \, \mathbf{1}_{q=p}, \tag{20}$$

where the $[\, \Sigma_f(\cdot) \,]_{qp}$ denotes the $q$-th row and $p$-th column element of $\Sigma_f(\cdot) \in R^{Q \times Q}$, and the $\mathrm{vec}(\cdot)$ denotes the operator that maps the matrix $\cdot$ to the corresponding vector by concatenating the row vectors of the matrix. The $\mathbf{1}_{q=p}$ denotes a indicator function. As a result, the function space covariance is obtained without using the automatic differentiation, as follows:

$$\Sigma_f(\cdot) = \mathrm{diag}\left(\left[\, \|h(\cdot)\|^2_{\sigma^2_1} \,, \,\ldots\,, \,\, \|h(\cdot)\|^2_{\sigma^2_Q} \,\right]\right) \in R^{Q \times Q}, \tag{21}$$

This closed form of the variance in Eq. (21) enables the proposed inference to build the sample function $\{f_{(j)}(\cdot)\}^j_{j=1}$ by applying the reparameterization trick on function space as described in Eq. (15) as well as to use the regularization of the KL divergence on function space in Eqs. (5) and (6) without the use of Jacobian computation.

### A.3 INTERPRETATION OF THE PROPOSED FUNCTION SPACE PRIOR

The proposed function prior $p_{\mathrm{BMA}}(\cdot)$, defined by the following parameters $\mu_{f_p}(\cdot)$ and $\Sigma_{f_p}(\cdot)$ of Eq. (11), can be understood as to have the following predictive probability $\hat{p}_{\mathrm{BMA}}(\cdot)$ via MPA approximation:

$$\hat{p}_{\mathrm{BMA}}(\cdot) := \int \mathrm{Softmax}(f_{\mathrm{BMA}}(\cdot)) \, \mathcal{N}\big(f_{\mathrm{BMA}}(\cdot); \mu_{f_p}(\cdot) \,, \, \Sigma_{f_p}(\cdot)\big) df_{\mathrm{BMA}} \tag{22}$$

$$\approx \mathrm{Softmax}\left(\mu_{f_p}(\cdot) \big/ \sqrt{1 + \pi/8 \Sigma_{f_p}(\cdot)}\right). \tag{23}$$

When the context input $\cdot \in X_{\mathcal{I}}$ is close to IND-like set, the $\mathrm{proj}\big(h(\cdot)\big)$ would be projected to one element out of $\{\hat{m}^q_h\}^Q_{q=1}$. Then, since the empirical weight parameters $(\hat{\mu}^L_w, \hat{\Sigma}^L_w)$ of Eq. (9) are obtained at the pre-determined stages of training that can incorporate the inductive bias of training set $\mathcal{D}$, the $\mu_{f_p}(\cdot)$ has the high peaked logits and $\Sigma_{f_p}(\cdot)$ has the small variance. This allows $\hat{p}_{\mathrm{BMA}}(\cdot)$ of Eq. (23) to have the high peaked logits (small entropy).

# B  APPENDIX:EXPERIMENT DETAILS

## B.1  HYPERPARAMETERS SETTING OF THE PROPOSED INFERENCE

We basically follow the well-known training hyperparameters configurations in (He et al., 2016; Dosovitskiy et al., 2021).

For the training of the ResNet for CIFAR 10 and 100, we set 200 epochs, 128 batch size, and use SGD optimizer with $10^{-1}$ learning rate, $5 \times 10^{-4}$ weight decay, and $0.9$ momentum. We employ the cosine learning scheduler after 10 warm-up epochs.

For the training of the VIT for CIFAR 100, we set 10000 steps, 512 batch size (8 step gradient accumulation with 64 batch size), and use SGD optimizer with $3 \times 10^{-2}$ learning rate and $0.9$ momentum. We employ the cosine learning scheduler after 500 warm-up steps as following the training scheme in the appendix of (Dosovitskiy et al., 2021).

For the proposed inference, we use the following hyperparameters in Table 3.

| Dataset | Hyperparameters | Range |
|---|---|---|
| All | Variance of variational weight parameters $\log \sigma_{w_q}$ | $\mathcal{U}(-6, -5)$ |
|  | KL regularization $\lambda$ in Eqs. (1) and (2) | $\{10^{-3}, 10^{-4}, 10^{-5}, 10^{-6}\}$ |
|  | The number of context inputs per batch | 32 / 128 (ResNet), 16/64 (VIT) |
|  | Pre-determined iterations $\mathcal{T}$ with $T$ iterations | $0.8T + \{-20, -16, -12, -8, -4\}$ |
|  | The number of sample functions $J$ in Eq. (7) | 1 (training), 10 (validation) |
|  | Radius $r$ in Eq. (12) for adversarial hidden feature | $\{0.01, 0.05, 0.10\}$ |
|  | Temperature $\tau$ in Eq. (15) for peaked logits | $\{1/0.9 , 1/0.75 , 1/0.5\}$ |
| CIFAR 10 | Top-K for latent variable $z(\cdot)$ | 3 |
| CIFAR 100 | Top-K for latent variable $z(\cdot)$ | $\{5, 10\}$ |

Table 3: Hyperparameters settings of the proposed inference (R-FVI).

For the other inferences, we use the following hyperparameters in Table 4.

| Inference | Hyperparameters | Range |
|---|---|---|
| MAP | MAP regularization hyperparameter $\lambda$ | $\{10^{-2}, 10^{-3}\}$ |
| MFVI | Mean of prior weight parameters $\mu_{w_p}$ | 0 |
|  | Variance of prior weight parameters $\log \sigma_{w_p}$ | $\mathcal{U}(0, 1)$ |
|  | Variance of variational weight parameters $\log \sigma_{w_q}$ | $\mathcal{U}(-6, -5)$ |
|  | KL regularization $\lambda$ in Eq. (1) | $\{10^{-4}, 10^{-5}, 10^{-6}\}$ |
|  | The number of sample functions $J$ in Eq. (7) | 1 (training), 10 (validation) |
| T-FVI | Mean of function space prior $\mu_{f_p}$ | 0 |
|  | Variance of function space prior $\sigma_{f_p}$ | $\{1, 5, 10, 100\}$ |
|  | Variance of variational weight parameters $\log \sigma_{w_q}$ | $\mathcal{U}(-6, -5)$ |
|  | KL regularization $\lambda$ in Eq. (2) | $\{10^{-4}, 10^{-5}, 10^{-6}\}$ |
|  | The number of context inputs per batch | 32 / 128 (ResNet), 16/64 (VIT) |
|  | The number of sample functions $J$ in Eq. (7) | 1 (training), 10 (validation) |

Table 4: Hyperparameters settings of the baseline inferences.

## B.2 ADDITIONAL EXPERIMENT RESULTS FOR SECTION 5.1

We consider the various context inputs to investigate how uniform function space prior affects the performances of the function space BNNs depending on the similarity between the training inputs and context inputs. Specifically, we set CIFAR 10 as training set and CIFAR 100 as additional inputs, and set the context inputs $(1 - \alpha)x_{\mathrm{tr}} + \alpha x_{\mathrm{add}}$ by mixing the training input $x_{\mathrm{tr}}$ and additional input $x_{\mathrm{add}}$ at the mixing level $\alpha \in \{.2, .5, .8\}$. Then, we train the ResNet 20 by using the f-EBLO $\mathcal{L}(\Theta_f)$ in Eq. (2) along with the uniform function space prior distribution $p(f_{\mathrm{lin}}(\cdot)) = N(f_{\mathrm{lin}}(\cdot); 0, 10I)$.

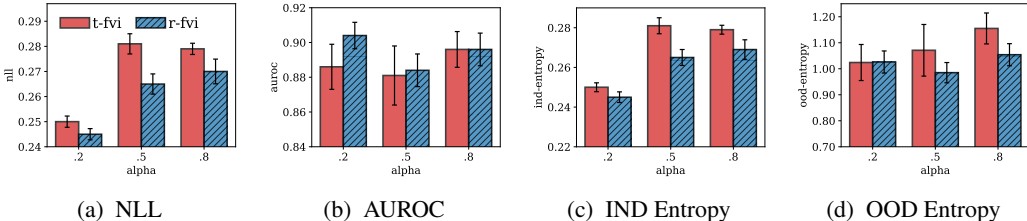

     (a) NLL           (b) AUROC          (c) IND Entropy      (d) OOD Entropy

Figure 5: Effect of similarity between the training sets and the context inputs; we run 5 experiments with different random seeds and report the mean and one-standard error of each metric. **Panel (a)** shows the test log likelihood on CIFAR 10 over different hyperparameter $\alpha$ controlling the similarity between the training and the context inputs. **Panel (b)** shows the corresponding AUROC on SVHN dataset. **Panel (c)** and **Panel (d)** show the predictive entropy on CIFAR 10 and SVHN, respectively.

Fig. 5 compares the results of the baseline inference (T-FVI) and proposed inference (R-FVI) when setting the regularization hyperparameter $\lambda = 1e^{-3}$. We see that as the context inputs, far from the training inputs ($\alpha \in \{.5, .8\}$) are used, the NLL of the baseline degrades, whereas its AUROC improves. Similarly, the values of predictive entropy evaluated on CIFAR 10 (IND) and SVHN (OOD) increase as shown in each left side of Figs. 5c and 5d. Similarly, the proposed approach shows a similar trend with the those of baseline.

From these results, we guess that the similarity between the training inputs and context inputs seems to affect the performances of the function space BNNs, and using the context inputs, that are close to the training inputs, would be helpful to build a effective function space prior.

## B.3 ADDITIONAL EXPERIMENT RESULTS FOR SECTION 5.2

**Details of Experimental setup.** Due to the results of Fig. 5, we observe that using the context input directly to build the function space prior would degrade the performance of NLL and AUROC. Therefore, after searching $\alpha \in (0, 1)$, we set $a = 0.1$ and use the context input as $(1 - \alpha)x_{\mathrm{tr}} + \alpha x_{\mathrm{add}}$ by mixing the training input $x_{\mathrm{tr}}$ and additional input $x_{\mathrm{add}} \in \{\mathrm{CIFAR\ 100}, \mathrm{Tiny\text{-}ImageNet}\}$.

Additionally, we observe that although the training the model by the existing variational inference (MFVI and T-FVI) with the cold posterior effect, i.e., $\lambda \in (0, 1)$ are more effective than using $\lambda = 1$, employing the MAP inference for the first training stages and then applying the variational inference for the left iterations results in better performance compared to those of the variational inference with cold posterior effect. Therefore, for the training of MFVI and T-FVI, we first apply the MAP inference until the last iteration $t_T$ of the pre-determined iterations $\mathcal{T} = \{t_1, .., t_T\}$, and report the corresponding result in this work; note that the period of MAP inference for MFVI and T-FVI are same with that of the proposed inference (R-FVI).

| Model/Data | Method | Context set | ACC ↑ | NLL ↓ | ECE ↓ | AUROC ↑ |
|---|---|---|---|---|---|---|
| ResNet 20 CIFAR 10 | MAP | | (**0.923**, 0.000) | (**0.236**, 0.004) | (0.010, 0.001) | (0.860, 0.029) |
| | MFVI | | (0.922, 0.001) | (0.240, 0.004) | (0.012, 0.000) | (0.868, 0.031) |
| | T-FSVI | CIFAR 100 | (0.923, 0.001) | (0.236, 0.002) | (0.010, 0.001) | (0.884, 0.026) |
| | R-FVI | CIFAR 100 | (0.921, 0.002) | (0.239, 0.002) | (**0.009**, 0.000) | (0.871, 0.009) |
| | | ADV-h | (**0.923**, 0.001) | (0.243, 0.002) | (0.012, 0.000) | (**0.896**, 0.005) |

Table 5: Result of the ResNet 20 on CIFAR 10.

**Result of ResNet 20 on CIFAR 10.** Table 5 compares the inference methods over different 3 seeds when training the ResNet 20 on CIFAR 10.

**Ablation study on the KL regularisation hyperparameter $\lambda$.** We investigate the effect of the KL regularisation hyperparameter $\lambda$ for the proposed inference. In this procedure, we use ResNet 18 with CIFAR 100 set. We consider the varying $\lambda \in \{1e^{-4}, 1e^{-5}, 1e^{-6}\}$, and unify the condition of the other hyperparameters. Table 6 compares the effect of KL regularisation hyperparameter over 3 seeds.

| Method | Context set | ACC ↑ | NLL ↓ | ECE ↓ | AUROC ↑ |
|---|---|---|---|---|---|
| $\boldsymbol{\lambda = 1e^{-4}}, \tau = 1/0.9$ | | | | | |
| R-FVI ($J = 10$) | T-ImageNet | (0.775, 0.001) | (0.905, 0.002) | (0.045, 0.004) | (0.879, 0.013) |
| R-FVI ($J = 100$) | | (0.775, 0.000) | (0.903, 0.003) | (0.045, 0.002) | (0.880, 0.011) |
| $\boldsymbol{\lambda = 1e^{-5}}, \tau = 1/0.9$ | | | | | |
| R-FVI ($J = 10$) | T-ImageNet | (0.775, 0.000) | (0.913, 0.002) | (0.047, 0.003) | (0.886, 0.013) |
| R-FVI ($J = 100$) | | (0.776, 0.001) | (0.912, 0.003) | (0.046, 0.004) | (0.886, 0.012) |
| $\boldsymbol{\lambda = 1e^{-6}}, \tau = 1/0.9$ | | | | | |
| R-FVI ($J = 10$) | T-ImageNet | (0.775, 0.001) | (0.915, 0.003) | (0.046, 0.003) | (0.885, 0.014) |
| R-FVI ($J = 100$) | | (0.776, 0.001) | (0.913, 0.003) | (0.046, 0.004) | (0.885, 0.013) |

Table 6: Effect of the KL regularization hyperparameter $\lambda$

**Ablation study on the radius $r$.** We investigate the effect of the radius for the adversarial context hidden features. In this procedure, we use ResNet 18 with CIFAR 10 set. We consider the varying $r \in \{0.01, 0.05\}$ and the random radius $r \in U(0.1, 0.2)$, and unify the condition of the other hyperparameters. Table 7 compares the effect of the radius over different 3 seeds.

| Method | Context set | ACC ↑ | NLL ↓ | ECE ↓ | AUROC ↑ |
|---|---|---|---|---|---|
| $\boldsymbol{r = 0.01}, \lambda = 1e^{-3}, \tau = 1/0.9$ | | | | | |
| R-FVI ($J = 10$) | ADV-h | (0.951, 0.000) | (0.168, 0.001) | (0.012, 0.000) | (0.952, 0.006) |
| R-FVI ($J = 100$) | | (0.951, 0.001) | (0.168, 0.001) | (0.011, 0.000) | (0.952, 0.006) |
| $\boldsymbol{r = 0.05}, \lambda = 1e^{-3}, \tau = 1/0.9$ | | | | | |
| R-FVI ($J = 10$) | ADV-h | (0.951, 0.000) | (0.167, 0.001) | (0.011, 0.000) | (0.956, 0.002) |
| R-FVI ($J = 100$) | | (0.951, 0.000) | (0.167, 0.001) | (0.010, 0.001) | (0.957, 0.002) |
| $\boldsymbol{r \in U(0.10, 0.20)}, \lambda = 1e^{-3}, \tau = 1/0.9$ | | | | | |
| R-FVI ($J = 10$) | ADV-h | (0.951, 0.001) | (0.173, 0.003) | (0.011, 0.0) | (0.949, 0.005) |
| R-FVI ($J = 100$) | | (0.951, 0.001) | (0.172, 0.003) | (0.010, 0.001) | (0.951, 0.005) |

Table 7: Effect of the radius $r$ of adversarial context hidden feature

**Ablation study on the temperature hyperparameter $\tau$.** We investigate the effect of the temperature hyperparameter $\tau$ for the peaked logits value. In this procedure, we use ResNet 18 with CIFAR 10 using the additional context input as CIFAR 100, and adversarial hidden feature. In this procedure, we consider the varying $\tau \in \{1/0.9, 1/0.75, 1/0.5\}$, and unify the condition of the other hyperparameters. Table 8 compares the effect of temperature hyperparameter over different 3 seeds.

| Method | Context set | ACC ↑ | NLL ↓ | ECE ↓ | AUROC ↑ |
|---|---|---|---|---|---|
| $\boldsymbol{\tau = 1/0.9}, \lambda = 1e^{-5}$ (CIFAR100), $\lambda = 1e^{-3}$ (ADV-h), $r = 0.05$ | | | | | |
| R-FVI ($J = 10$) | CIFAR 100 | (0.950, 0.001) | (0.173, 0.002) | (0.010, 0.000) | (0.940, 0.005) |
| R-FVI ($J = 100$) | | (0.951, 0.000) | (0.171, 0.002) | (0.009, 0.000) | (0.942, 0.005) |
| R-FVI ($J = 10$) | ADV-h | (0.951, 0.000) | (0.167, 0.001) | (0.011, 0.000) | (0.956, 0.002) |
| R-FVI ($J = 100$) | | (0.951, 0.000) | (0.167, 0.001) | (0.010, 0.001) | (0.957, 0.002) |
| $\boldsymbol{\tau = 1/0.75}, \lambda = 1e^{-5}$ (CIFAR100), $\lambda = 1e^{-3}$ (ADV-h), $r = 0.05$ | | | | | |
| R-FVI ($J = 10$) | CIFAR 100 | (0.951, 0.000) | (0.168, 0.002) | (0.010, 0.001) | (0.945, 0.008) |
| R-FVI ($J = 100$) | | (0.951, 0.001) | (0.168, 0.002) | (0.010, 0.000) | (0.945, 0.008) |
| R-FVI ($J = 10$) | ADV-h | (0.952, 0.001) | (0.169, 0.000) | (0.010, 0.001) | (0.951, 0.003) |
| R-FVI ($J = 100$) | | (0.952, 0.001) | (0.168, 0.000) | (0.010, 0.000) | (0.953, 0.002) |
| $\boldsymbol{\tau = 1/0.5}, \lambda = 1e^{-5}$ (CIFAR100), $\lambda = 1e^{-3}$ (ADV-h), $r = 0.05$ | | | | | |
| R-FVI ($J = 10$) | CIFAR 100 | (0.950, 0.001) | (0.169, 0.003) | (0.011, 0.000) | (0.956, 0.003) |
| R-FVI ($J = 100$) | | (0.950, 0.001) | (0.169, 0.003) | (0.011, 0.000) | (0.955, 0.003) |
| R-FVI ($J = 10$) | ADV-h* | 0.949 | 0.177 | 0.010 | 0.943 |
| R-FVI ($J = 100$) | | 0.950 | 0.176 | 0.010 | 0.945 |

Table 8: Effect of the temperature hyperparameter: the result of * is obtained with one random seed.

**Ablation study on the top-k marginalized distribution of $z(\cdot)$.** We investigate the effect of the top-k marginalized distribution $z(\cdot)$ for the high dimensional output. In this procedure, we use ResNet 18 with CIFAR 100 using the adversarial hidden feature. In this procedure, we consider the varying $\text{top-k} \in \{3, 5, 10, 20\}$, and unify the condition of the other hyperparameters. Table 9 compares the effect of the top-k marginalized distribution over different 3 seeds.

| Method | Context set | ACC ↑ | NLL ↓ | ECE ↓ | AUROC ↑ |
|---|---|---|---|---|---|
| $\textbf{Top-k} = 3, \lambda = 1e^{-3}, \tau = 0.75, r = 0.1$ | | | | | |
| R-FVI ($J = 10$) | ADV-h | (0.778, 0.002) | (0.904, 0.001) | (0.047, 0.002) | (0.876, 0.017) |
| R-FVI ($J = 100$) | | (0.779, 0.001) | (0.898, 0.001) | (0.046, 0.001) | (0.876, 0.016) |
| $\textbf{Top-k} = 5, \lambda = 1e^{-3}, \tau = 0.75, r = 0.1$ | | | | | |
| R-FVI ($J = 10$) | ADV-h | (0.779, 0.001) | (0.893, 0.002) | (0.047, 0.002) | (0.877, 0.019) |
| R-FVI ($J = 100$) | | (0.780, 0.001) | (0.888, 0.002) | (0.047, 0.003) | (0.877, 0.018) |
| $\textbf{Top-k} = 10, \lambda = 1e^{-3}, \tau = 0.75, r = 0.1$ | | | | | |
| R-FVI ($J = 10$) | ADV-h | (0.778, 0.001) | (0.894, 0.003) | (0.047, 0.004) | (0.883, 0.014) |
| R-FVI ($J = 100$) | | (0.778, 0.001) | (0.894, 0.003) | (0.047, 0.004) | (0.883, 0.014) |
| $\textbf{Top-k} = 20, \lambda = 1e^{-3}, \tau = 0.75, r = 0.1$ | | | | | |
| R-FVI ($J = 10$) | ADV-h | (0.778, 0.002) | (0.904, 0.002) | (0.050, 0.005) | (0.878, 0.017) |
| R-FVI ($J = 100$) | | (0.778, 0.002) | (0.901, 0.002) | (0.049, 0.006) | (0.878, 0.017) |

Table 9: Investigation of the top-k marginalized distribution of $z(\cdot)$ on CIFAR 100.

### B.4 ADDITIONAL EXPERIMENT RESULTS FOR SECTION 5.3

We validate the proposed inference by evaluating the uncertainty estimation performance on the corrupted CIFAR 10 (Hendrycks & Dietterich, 2018), having the different types of covariate shift and each shift set consists of 5 intensity levels. We compare the ResNet 18 trained by the proposed inference method (R-FVI) with the adversarial hidden feature to the baseline inference (T-FVI) for CIFAR 1, which are reported in Table 1.

**Results.** Figs. 6 and 7 compare the accuracy and calibrated NLL of the proposed inference and and baseline inference, respectively. We that the proposed inference (R-FVI) tends to have the higher accuracy and smaller NLL values in the datasets of damaged intensities 1, 2, and 3.

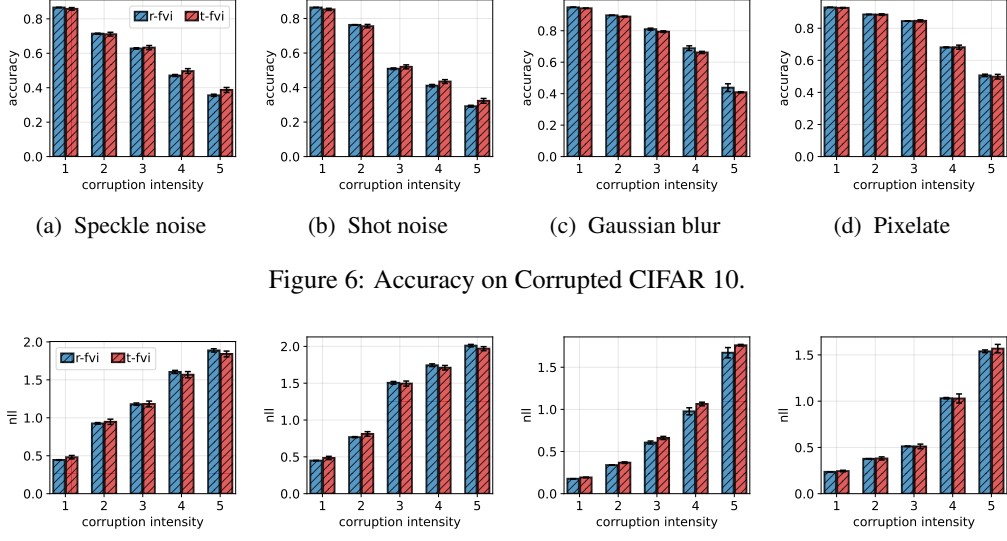

|  (a)  Speckle noise | (b)  Shot noise | (c)  Gaussian blur | (d)  Pixelate |

Figure 6: Accuracy on Corrupted CIFAR 10.

|  (a)  Speckle noise | (b)  Shot noise | (c)  Gaussian blur | (d)  Pixelate |

Figure 7: NLL on Corrupted CIFAR 10.

