# OpenReview forum: "Improved Function Space Variational Inference with Informative Priors"
_ICLR.cc/2024/Conference — Submitted to ICLR 2024_

### Official Review · Reviewer_QnS6 · 2023-10-29

**Soundness:** 3 good
**Presentation:** 3 good
**Contribution:** 3 good
**Rating:** 6
**Confidence:** 3

**Summary:**

This paper proposes a new function space variational inference for classification problem. The existing informative function prior may lead to high entropy to both in and out distribution data points. The idea is to assign lower entropy to in-distribution data and higher entropy to out-of-distribution data. The authors have designed a specific function prior and variational posterior to control the entropies. The experiments show somewhat promising results.

**Strengths:**

The paper is well presented with good visualization to motivate the targeted problem. The designed prior and variational distribution are interesting and reasonable.

**Weaknesses:**

1.	Why was the last layer set as BNN layer? Why not set the whole network as BNN?
2.	The experimental results are only marginal, which is my main concern. Is that because the only last layer is BNN? Why not try more BNN layers that can demonstrate more difference between new prior with previous informative prior?
3.	Since the prior is changing during the training, how to ensure the convergence of the procedure?
4.	Since the method is specially designed for the classification task, I suggest the author to revise the title and introduction accordingly to highlight the classification task.
5.	What is the role of (12)?
6.	Some symbols are not defined, like N^q

**Questions:**

Please see the Weaknesses.

---

> ### Author Response · Authors · 2023-11-19
> **Responses to Reviewer QnS6**
>
> ### Q1. Why was the last layer set as BNN layer? Why not set the whole network as BNN?
>
> > We request you to confirm our response regarding the **Reasons for taking the Last-layer BNNs** in the general review.
>
> ### Q2. The experimental results are only marginal. Is that because the only last layer is BNN? Why not try more BNN layers that can demonstrate more difference between new prior with previous informative prior?
>
> > We request you to confirm our response regarding the **Marginal Performance improvement, possibly due to the Last-layer BNN** in the general review.
>
>
> ### Q3. Since the prior is changing during the training, how to ensure the convergence of the procedure?
>
> > Our function-space prior does not vary during training. Please refer to our response in the **Procedure of the proposed algorithm** above and Algorithm 1 on page 12 for details.
>
>
> ### Q4. Since the method is specially designed for the classification task, I suggest the author revise the title and introduction accordingly to highlight the classification task.
>
> > We have extended our method for the regression task and included regression results. Please see our response in the **Extension for regression tasks**. We will reflect this in the final manuscript.
>
>
>
>
>
> ### Q5. What is the role of Eq. (12)?
>
> > The function-space prior requires auxiliary inputs, as discussed in our response regarding the **distinct property of the function-space prior as an input-dependent prior**. However, selecting additional datasets for the auxiliary inputs could be cumbersome if those datasets are not readily available. Therefore, for each training input $x_i$, we consider the $h(\cdot)$ from Eq. (12) to be used for constructing the function-space prior. It's important to note that the $h(\cdot)$ from Eq. (12) using $x_i$ is different from $h(x_i)$.  Therefor, this results in a function-space prior derived from $h(\cdot)$ of Eq. (12), which is distinct from the one derived from $h(x_i)$.
>
>
>
> ### Q6. Some symbols are not defined, like N^q
>
> > The $N^q$ represents the number of the labels corresponding to $q$ for the training set, i.e., $N^q= \vert \ \\{y_n \ | \, y_n=q, y_n \in \mathcal{D}\\} \ \vert$, for the training set $\mathcal{D}=\\{(x_n, y_n)\\}_{n=1}^{N}$ and $y_n \in \\{1,..,Q\\}$. For other errors, we will correct them in the final manuscript.

---

> > ### Author Response · Authors · 2023-11-22
> > **Official comment by authors**
> >
> > We kindly remind you that the discussion period will be closing soon. Therefore, we kindly ask for your response to our reply if you have any further concerns or questions regarding our work. Alternatively, we kindly request that you re-evaluate our work and consider raising our score.

---

> > ### Comment · Reviewer_QnS6 · 2023-11-22
> >
> > Thanks for the authors' responses!
> >
> > I understand the difficulty of adopting full BNNs for large networks, but the authors could at least try 3 to 10 BNN layers rather than one. The results in the explanation of the marginal improvement are still marginal to me. The extension to regression is weird to me. In order to use the proposed method, the authors have to discretize the continuous outputs. I can understand why the entropies work for classification tasks, but I do not understand why there are classification entropy properties in continuous outputs. Hence, I am going to keep my socre.

---

### Official Review · Reviewer_hyeW · 2023-10-30

**Soundness:** 1 poor
**Presentation:** 1 poor
**Contribution:** 1 poor
**Rating:** 3
**Confidence:** 4

**Summary:**

The paper studies the problem of inferring posteriors of Bayesian neural nets with function-space priors more effectively than the existing variational inference approach that uses the first-order Taylor approximation.

**Strengths:**

* The studied transfer learning setup with Vision Transformers is interesting and fits well to the purpose of doing Bayesian inference.

 * The paper presents results from a comprehensive set of experiments.

**Weaknesses:**

* The problem setup does not make much sense to me. The introduction says:   **“We build an informative function space prior by using the empirical Bayes approach along with the parameters of hidden values and the last layer weight parameters which are obtained iterations during early stages of training”**. I wonder how the approach is then different from having an uninformative prior after all. If the information comes from the data, it is technically not a prior. It appears that the paper makes its main point by differentiating from cases where the priors are just so strong that they unnecessarily restrict the model capacity.

 * The paper significantly lacks clarity. Apart from having extremely many typographical errors, it has statements without sharp enough meanings. Among the many, one example is: **“Denote auxiliary inputs, which are far from training points and are placed closely with the training sets, respectively”**. I have no idea what it means for a training point to be close to a training set. Likewise: **“h(.) from the q-th component empirical parameters of hidden feature …”** What is an empirical parameter? I also have no clue about what is going on in pages 5 and 6 after spending considerable time trying to read them. Even the purpose of all these complications such as introducing adversarial hidden features do not look to me justified.

 * It is a clear weakness that after motivating Bayesian inference with lots of effort, the paper ends up using it only on the penultimate layer. Those layers are typically linear, where even closed-form Bayesian linear regression would work and the learned model weights would give a degree of interpretability. Why should one use function-space Bayesian inference if the prior will come from the weights learned in another data set and only the penultimate layer will be Bayesianized?

 * I do not think the reported results demonstrate the benefit of the proposed approach clearly enough. All models in all experiments perform very closely to each other. The results reported in Figure 2 are mixed: (a) and © are favorable for the central message of the paper while (b) and (d) are just the opposite.

* The take-home message given in the last sentence of Section 5.1 is obvious and comes from the nature of using an arbitrary regularizer. I wonder why one needs even an experiment for that.

--- POST REBUTTAL ---

The author response does not give any concrete answer to any of the issues I raised above. I keep my score unchanged.

**Questions:**

Only T-FVI or all the three baselines? If first, why not others?

---

> ### Author Response · Authors · 2023-11-19
> **Responses to Reviewer hyeW**
>
> ### Q1. I wonder how the approach is then different from having an uninformative prior after all. If the information comes from the data, it is technically not a prior.
>
> > The proposed function-space prior is adaptively constructed based on auxiliary inputs, as described in **The Distinct Property of the Function-Space Prior as an Input-Dependent Prior**. We devise the function-space prior in such a way that if the auxiliary inputs appear more similar to the in-distribution dataset, the proposed function-space prior would exhibit higher a peaked probability due to its construction, as outlined in the initial phase of the **Procedure of the proposed algorithm**. The regularization of this prior would adaptively influence the model parameters depending on the auxiliary input (for example, $A$: the subset of CIFAR 100 sharing the label of CIFAR 10, and $B$: the remaining datasets of CIFAR 100). This is definitely distinct to the uninformative prior applying the same regularization to both $A$ and $B$.
>
> > Regarding the comment about the prior using the dataset, we partially agree with it because the empirical Bayesian method [1] sets the parameters of the prior distribution by training on the datasets. We adopt a similar approach to establish the function-space prior.
>
> > [1] https://en.wikipedia.org/wiki/Empirical_Bayes_method
>
>
> ###  Q2. “Denote auxiliary input, I have no idea what it means for a training point to be close to a training set. Likewise: “h(.) from the q-th component empirical parameters of hidden features …” What is an empirical parameter?
>
> > In light of the feedback from other reviewers, we acknowledge that our manuscript seems difficult to understand. We kindly request you to confirm the responses from all reviewers above. Then, we recommend revisiting sections 2, 3, and 4 for a clearer understanding. Also, we will enhance the clarity of our manuscript for easier understanding.
>
>
> ### Q3. When applying the last-layer, the corresponding model would be Bayesian linear regression with interpretable parameters. Why should one use function-space Bayesian inference if the prior will come from the weights learned in another data set and only the penultimate layer will be Bayesianized?
>
> > Although our proposed approach models only the last layer to be the Bayesian layer, the regularization via the function-space KL divergence of Eq. (6) has an impact on the overall parameter learning, including the weight parameters of the $L-1$ deterministic layers $\\{f^{l} \\}_{l=1}^{L-1}$.
>
> > Note that from a Bayesian perspective, the weight parameters of  L-1 deterministic layers $\\{f^{l}\\}_{l=1}^{L-1}$ can be also understood as the mean parameters of the variational distribution for the BNN having a fixed variance (not trainable) with a very small value.
>
>
>
> ### Q4. The results reported in Figure 2 are mixed.  Figure 2.(a) and (C) are favorable for the central message of the paper while Figure 2.(b) and (d) are just the opposite.
>
> > We would like to emphasize that the primary message from Figure 2 is that the proposed method results in more reliable uncertainty estimation for both the in-training set (lower NLL) and out-of-training test set (higher AUROC) across various regularization parameters λ. Therefore, it demonstrates that the proposed method effectively mitigates the trade-off issue associated with the existing input-independent function-space prior.
>
>
> ### Q5. The take-home message given in the last sentence of Section 5.1 is obvious and comes from the nature of using an arbitrary regularizer. I wonder why one even needs an experiment for that.
>
> > We agree that controlling the regularization hyperparameter $\lambda$ can alleviate the trade-off issue for the input-independent prior. However, we propose a more effective solution by incorporating the input-dependent function-space prior. Note that the NLL and AUROC of the proposed method (R-FVI) consistently outperform the baseline (T-FVI) for all varying regularization hyperparameters in Figure 2 (a) and (b).
>
>
> ### Q6. Only T-FVI or all the three baselines? If first, why not others?
>
> > Since we aim at tackling the issue of input-independent function-space prior for T-FVI in this work, we set the T-FVI as our main baseline.

---

> > ### Author Response · Authors · 2023-11-22
> > **Official comment by authors**
> >
> > We kindly remind you that the discussion period will be closing soon. Therefore, we kindly ask for your response to our reply if you have any further concerns or questions regarding our work. Alternatively, we kindly request that you re-evaluate our work and consider raising our score.

---

### Official Review · Reviewer_3j4p · 2023-10-30

**Soundness:** 1 poor
**Presentation:** 2 fair
**Contribution:** 2 fair
**Rating:** 3
**Confidence:** 3

**Summary:**

This paper investigates a new approach to specify informative priors for improve variational inference in function space on classification tasks. In particular, the paper relies on the functional variational inference (f-VI) framework proposed by Rudner et al. (2022) but replaces the uniform (uninformative) functional prior with an informative functional prior. To this end, the authors reconsider the role of functional prior in the perspective of Bayesian Model Averaging (BMA), and then propose a new functional prior relying on the empirical Bayes approach (using the training data to specify the prior). This prior is aimed at avoiding common pathology of the uniform functional prior, which encourages the model to be uncertain on both the training and out-of-training data. In addition, the authors propose a new functional variational distribution that is aligned with this new functional prior.  The proposed method is validated on toy data and popular benchmarks.

**Strengths:**

- The paper aims at tackling an important problem for Bayesian deep learning which is designing a good prior for Bayesian neural networks.
- The code is anonymously provided. However, there are no instructions to use the code. Thus, it is difficult to verify the provided code.
- The idea is interesting and well-motivated which is to design a new prior promoting high uncertainty for out-of-distribution data but low uncertainty for in-distribution data.

**Weaknesses:**

- The writing should be improved. There are many grammar typos such as the use of “a” and “an”. Some parts of the paper are difficult to read, especially Section 4. There are some confusions of notations. Please refer these to in the box of Questions.
- In Section 4, although the authors motivated the paper from the view of Bayesian Model Averaging, and claimed that *“we may design a function space prior that does not explicitly encourage generating high-entropy predictions for each predictive probability but the average prediction (via BMA) would still have high-entropy when encountered with an OOD input”*, it is not clear how the proposed prior can achieve this.
- The proposed method is somehow ad-hoc without well-elaborations. For example, in Eq (9), why do the empirical mean and covariance are averaged from those obtained from all pre-training iterations? How did the authors come up with the equation (13), the parameter $\hat{p}(\cdot)$ for the variational distribution?
- The authors ignored a very related work from Izmailov et al. (2021). The narrative of this work also relies on Bayesian model averaging. This work also considers designing a novel prior that is robust to out-of-distribution data by using the empirical Bayes approach. The authors should cite, discuss, and compare experimentally with this work.
- Experimental results on image benchmarks (Sec 5.2) show that the performance gain from the proposed prior (R-FVI) is very marginal compared to the uniform prior (T-FVI). To show clearly the effect of the prior, the authors should ablate different training sizes and temperature values for the posterior on these benchmarks.

References:

Izmailov et al. Dangers of Bayesian Model Averaging under Covariate Shift. NeurIPS 2021.

**Questions:**

- In Equation (8), what is $N^q$ how to define it? Do we have to compute the means and shared covariance over the dataset?
- In Equation (9): what is $T$? It should be consistent with the sentence before the Equation (8).
- From Figures 3c and 3f, it seems that the proposed method always induces a much higher disagreement ratio on both in- and out-of-distribution data compared to the uniform prior. Is this good? On in-distribution data, the entropy should be low.
- In the paragraph "Context inputs from adversarial hidden features". How do we define $r$?

---

> ### Author Response · Authors · 2023-11-19
> **Responses to Reviewer 3j4p**
>
> ### Q1. Regarding our claim that the proposed function-space prior allows the average prediction (via BMA) to have high-entropy for an OOD input, it is not clear  how the proposed function-space prior achieves this.
>
> > We request you to confirm our response regarding the **How the proposed methods achieve the BMA property using the inductive bias** in the general review.
>
>
> ### Q2.  Our method is somehow ad-hoc without well-elaboration for Eq (9) and Eq (13).
>
> > We request you to confirm our response regarding the **Elaboration of the proposed algorithm** in the general review.
>
>
> ### Q3. The authors ignored a very related work from Izmailov et al. (2021) relying on Bayesian model averaging and a novel prior.
>
> > Based on our understanding, in the referenced study [1], the authors addressed the issue of poor uncertainty estimation in BNNs on the covariate shift dataset when the BNNs, using a zero-mean Gaussian prior distribution, are trained using SGHMC (oracle Bayesian inference yielding an almost true posterior). To address this limitation, they identified shortcomings in the posterior distribution arising from the zero-mean Gaussian prior and proposed a novel empirical covariance for the prior distribution of weight parameters.
>
> >  We acknowledge some similarities with [1], where both approaches use training datasets to construct the prior distribution for BNNs with the aim of improving uncertainty estimation. However, our work specifically focuses on resolving the trade-off issue caused by the input-independent function-space prior. Therefore, we refine the function-space prior and variational distribution to achieve reliable uncertainty estimation for both the in-training distribution and out-of-distribution dataset. This objective distinguishes our work from [1], which primarily concentrated on addressing the covariate shift dataset. We will elaborate on this discussion in the main manuscript.
>
> >  [1] Dangers of Bayesian Model Averaging under Covariate Shift - NeurIPS 21
>
>
> ### Q4. Regarding the marginal performance gain in Section 5.2.
>
> > We request you to confirm our response in the **Marginal Performance improvement, possibly due to the Last-layer BNN**
>
>
>
>
> ### Q5. What is $N^q$, how to define it?,
>
> > The $N^q$ represents the number of the labels corresponding to $q$ for the training set, i.e., $N^q= \vert \ \\{y_n \ | \ y_n=q, y_n \in \mathcal{D} \\} \ \vert$, for the training set $\mathcal{D} =\\{(x_n, y_n)\\}_{n=1}^{N}$ and $y_n \in \\{1,..,Q\\}$.
>
> ### Q6. Do we have to compute the means and shared covariance over the dataset for Eq. (8)?
>
> > We can utilize a subset of the full training set for computing the mean and shared covariance. However, this introduces another consideration—specifically, how to choose the subset of the training set. If the subset is selected in a way that introduces bias, it can give rise to additional issues for the function-space prior and the latent variable $z$ in the variational distribution. Consequently, we opt for using the entire training set. Since the means and shared covariances are updated in a batch manner, there is no need for additional iterations to compute the mean and shared covariance.
>
>
>
> ### Q7. what is $T$ ?,  It should be consistent with the sentence before the Equation (8).
>
> > The $T$ represents the number of pre-determined iterations. For example, if $\mathcal{T}=\\{145, 150, 155, 160 \\}$, then $T=4$. We apologize for the error in this notation. We will clarify it in the final manuscript.

---

> > ### Author Response · Authors · 2023-11-19
> > **Responses to Reviewer 3j4p**
> >
> > ### Q8. From Figures 3c and 3f, it seems that the proposed method always induces a much higher disagreement ratio on both in- and out-of-distribution data compared to the uniform prior. Is this good? On in-distribution data, the entropy should be low.
> >
> > > We acknowledge that in Figure 3c and 3f, our proposed method (R-FVI) exhibits a higher disagreement ratio on both in-distribution and out-of-distribution data compared to the uniform function-space prior (T-FVI).
> >
> > > However, the disagreement ratio of R-FVI on in-distribution data is significantly smaller than that on out-of-distribution data  in Figure 3c. In addition, in Figure 3.b (in-distribution data), the entropy of R-FVI becomes smaller compared to T-FVI as the number ($J$) of sample functions increases. Conversely, in Figure 3.e (out-of-distribution data), the entropy of R-FVI is larger than that of T-FVI. This suggests that the sample functions having a higher disagreement ratio can induce the smaller predictive entropy on in-distribution data.
> >
> > ### Q9. What is $\gamma$ ?, in the paragraph "Context inputs from adversarial hidden features". How do we define it ?
> >
> > > The $\gamma$ denotes the radius for the neighborhood set of $h(x_i)$, i.e.,
> > $\\{h \ | \   ||h - h(x_i)||_2 \leq r \\}$ for a training input $x_i$ .
> >
> > > We determine the value of $\gamma$ through validation. Specifically, we consider $\gamma$ within the set $\\{0.01, 0.05, 0.1, 0.2\\}$ and select the value that yields the best negative log-likelihood (NLL) from the validation set. Additional details can be found in Table 7 on Page 16.

---

> > > ### Author Response · Authors · 2023-11-22
> > > **Official comment by authors**
> > >
> > > We kindly remind you that the discussion period will be closing soon. Therefore, we kindly ask for your response to our reply if you have any further concerns or questions regarding our work. Alternatively, we kindly request that you re-evaluate our work and consider raising our score.

---

> > > > ### Comment · Reviewer_3j4p · 2023-11-22
> > > > **Official Comment by Reviewer 3j4p**
> > > >
> > > > Thanks the Authors for the rebuttal. However, the rebuttal did not address satisfactorily all my concerns. I believe that the manuscript should be improved significantly and needs another round of review. Therefore, I would keep the current score.

---

### Official Review · Reviewer_1KC1 · 2023-11-08

**Soundness:** 3 good
**Presentation:** 2 fair
**Contribution:** 2 fair
**Rating:** 6
**Confidence:** 3

**Summary:**

This paper focuses on improving function-space Bayesian Neural Networks (BNNs) by addressing some of the key challenges they face in terms of dealing with significative prior distributions. Applied in a classification setting, the authors propose an informative function space prior that encourages sample functions to have a certain predictive probability and varying degrees of disagreements based on input status. They also tackle the issue of computing KL divergences in function space by using an adversarial hidden feature and refining the variational function space distribution. Experimental results show that their approach outperforms other inference methods on the CIFAR 100 dataset, demonstrating its effectiveness for large-scale models.

**Strengths:**

* The topic is quite interesting from the point of view of enlarging the contributions to the function-space approach to modern probabilistic machine learning. The topic of using function-space BNNs is relevant and promising, and further research such as this is very welcome.
* The proposal for function-space variational distribution introduces a categorical latent variable that represents the uncertainty in using a specific feature based on its empirical distribution. This allows for better understanding and interpretation of the model's behavior.
* The authors use multi-dimensional probit approximation (MPA) to obtain an approximate marginalization over a Gaussian distribution for obtaining $\hat{p}(\cdot)$. This technique helps to efficiently compute and approximate complex distributions, making it feasible to implement this approach.

**Weaknesses:**

* The article writing does not contribute to the overall appreciation of the work being done and should be thoroughly revised. I strongly encourage the authors to do an integral check on the text for improvements. This is quite noticeable, even the abstract should be revised to correct typos and improve the overall text flow and comprehension. A lot more care and effort have to be put in this regard.
* The proposed method relies on a last-layer approximation, which is not thoroughly discussed enough. While this approach can be employed with the right arguments, the authors do not make the efforts necessary to justify this choice or the consequences it may entail in the proposed technique.
* While the paper presents an improved function-space variational inference method, it does not provide extensive evaluation results or comparisons against other existing methods on benchmark datasets or real-world applications to demonstrate its superiority over alternative approaches. I think stronger experimental work is needed to further motivate the usage of the proposed approach. The contribution is itself interesting, but further experimental results would bolster the proposal (e.g. regression experiments, applying this method to specify the prior in other function-space inference methods to check the potential improvements, etc.).
* The proposal made in the article is strictly limited to BNNs, while other methods such as the one in [1] or Rodríguez-Santana et al. (2022) can be seen as "generalist approaches" where the function-space formulation can be done for many other models (not just BNNs, which are only particular cases).

*Note:* I condition my review score on the fact that some of these issues get fixed in the final draft version. Otherwise, I may be inclined to lower the score.


(see "**Questions**" for the references)

**Questions:**

* In the initial paragraph of the introduction, when mentioning function-space BNNs I would include the reference [1].
* Why would you argue that the main goal of function-space BNNs is "directly assigning prior distributions to the outputs of neural networks"? I would argue this can be done without function-space formulation, and that the main interest lies directly on the properties of the function space itself. I would even argue that the approach does not necessarily become more "user-friendly" due to the difficulties intrinsic to function space. I would appreciate more insight on these points.
* Given the BMA approach and the nature of the contribution in Rodríguez-Santana et al. (2022), how do these relate to each other? I think further discussion here could improve to make a more comprehensive overall picture.
* Does the last-layer approximation play a role in the final performance metrics? What results are achieved if this restriction is not applied and instead a full Bayesian NN is used?

---
### **Notes:**

* As the authors mention: "(Flam-Shepherd et al., 2017; Karaletsos & Bui, 2020; Tran et al., 2022), it has been less clear to specify the interpretable function space prior for the classification task" I would expect this contribution to try to either expand on this formulation or present a general contribution for classification problems (although one also could say that works such as Tran et al. 2022 could serve to that purpose). The formulation of the article up to Section 3 makes the reader think the authors are presenting a general approach both for regression and classification, while in reality they only do the latter. Thus, I would encourage the authors to be clear with these intentions from the beginning. Moreover, since the conversion to regression does not seem too far off from the method present, I strongly
* Results for the presented method in table 2 are highlighted, while there are other methods in ECE and AUROC that are competitive (with equal performance). This should be corrected.

### **Minor corrections:**

* Please correct typos. Just in the abstract there are some of them, such as the capitalized "Recent", the "the this function space" sentence, "thought the uniform function space..." should be revised. Further examples can be found all through the text, such as "lineariztion" or "Jacobin matrix".
* Maintain consistency, e.g. if you are using "function-space" on the title I would expect to keep the "-" throughout the text. On the same line, remove the red-colored subindex in page 3 (unless you use justify its usage further). Also, there are some inconsistencies also in singular and plural expressions, such as "since the posterior distribution of the weight parameters p(Θ|D) are not tractable in general". Please, correct the text carefully.
* Reference for Rodríguez-Santana et al. (2022) is missing the \' in "í"

### **References:**
[1] Ma, C., Li, Y., and Hernández-Lobato, J. M. (2019). “Variational implicit processes”. In: International Conference on Machine Learning, pp. 4222–4233.

---

> ### Author Response · Authors · 2023-11-19
> **Responses to  Reviewer 1KC1**
>
> ### Q1.  Authors do not make the efforts necessary to justify this last-layer approximation.
>
> > We explain why the last-layer approximation was considered in **Reasons for taking the Last-layer BNN** in the responses to all reviewers. We will also describe this in our final manuscript as well.
>
> ### Q2. Does the last-layer approximation play a role in the final performance metrics? What results are achieved if this restriction is not applied and instead a full Bayesian NN is used ?
>
> > We request you to confirm our response in the **Marginal Performance improvement, possibly due to the Last-layer BNN**  in the responses to all reviewers above.
>
>
>
> ### Q3. Comment about insufficient extensive evaluation results (evaluation on regression task) in **Weakness**
>
> > We request you to confirm our response in the **Response to all reviewers for ""Extension for regression tasks""** above.
>
>
> ### Q4. Why would you argue that the main goal of function-space BNNs is "directly assigning prior distributions to the outputs of neural networks?
>
> > When considering one of the Bayesian perspectives, which involves incorporating prior knowledge into the model, we believe that providing a more directly interpretable function-space prior for BNNs is a desirable direction for practitioners. We acknowledge that this perspective can be subjective and thus would like to moderate our claim regarding the main goal. Additionally, we agree with your assertion that addressing the characteristics of the function space and overcoming the challenges of function-space inference are indeed meaningful directions for improving function-space inference.
>
> >However, if we focus solely on the characteristics of the function space and technical research, we might overlook the advantages of the Bayesian approach—especially its ability to enhance performance with limited data by leveraging prior knowledge. Therefore, we believe that research on BNNs that can incorporate intended prior knowledge is crucial. Exploring ways to make the function-space prior more user-friendly is the key direction in this context.
>
>
>
> ### Q5. Connection to generalistic approach; Given the BMA approach and the nature of the contribution in Rodríguez-Santana et al. (2022), how do these relate to each other?
>
> > Based on our understanding, Sparse Implicit Process (SIP) [1] primarily contributes to improving the scalability and flexibility of the Implicit Process (IP), known for constructing an implicit function-space distribution to model data. SIP achieves this improvement by incorporating the inducing input idea and adversarial Variational Inference. Our research, on the other hand, focuses on BNNs, considered as an example of IP. We aim to enhance function-space Variational Inference for BNNs, inspired by the ideal role of function-space predictive sample functions.
>
> > Regarding the connection to these works, we believe that these studies offer interesting insights into the construction of a function-space prior. SIP forwards Gaussian noise and data through a neural network to obtain random functions. It then uses the empirical distribution (averaged mean and covariance of the random functions) to form and learn the implicit prior distribution on function space. In contrast, our study utilizes the paths of weight parameters during training to construct an input-dependent function-space prior with different predictive entropy for in-distribution and out-of-distribution datasets. This difference seems to originate from the fact that, while SIP focuses more on obtaining the implicit function-space prior for flexible modeling, our work concentrates more on the adaptive function-space prior for Bayesian Model Averaging (BMA) prediction. This represents our perspective on the connection between our work and SIP. If there are different opinions, feel free to share them. It could contribute to a more comprehensive discussion of the papers.
>
> > [1] Function-space Inference with Sparse Implicit Processes - ICML 22

---

> > ### Author Response · Authors · 2023-11-22
> > **Official comment by authors**
> >
> > We kindly remind you that the discussion period will be closing soon. Therefore, we kindly ask for your response to our reply if you have any further concerns or questions regarding our work. Alternatively, we kindly request that you re-evaluate our work and consider raising our score.

---

> > > ### Comment · Reviewer_1KC1 · 2023-11-22
> > > **Reply to the authors**
> > >
> > > First of all, I wanted to thank the authors for the very detailed response and the amount of extra work put forth in the rebuttal process. I think the submitted work strongly benefits from these extra points and I deem the submission to be much more complete with the information available currently. I consider this updated version of the work a very different paper with respect to the original submission (and certainly, an improved version of it).
> > >
> > > The authors have addressed many of my questions and concerns in their comments, and I appreciate the clarifying statements for these points. On the other hand, I wish the authors conducted a more thorough experimental phase comparing to those other methods I mentioned in the review, namely, VIP and SIP, to better showcase the properties of the proposed method. In particular, the current method should beat generalist approaches in tasks related to BNNs, and that should also strengthen the experimental part of the submission and would provide further heuristic support for the approach. This same comment applies both to the classification and the regression tasks, on which the authors have much to gain from simple extensions of current work.
> > >
> > > In general, I am now more inclined than before to a positive review of the submitted work. However, given the interesting ideas and the amount of work already put forth in the submission, I consider the authors could provide a much-improved version with some very simple extensions and a bit of re-writing of the original text. Therefore, even though a bit conflicted by these facts, I  am inclined to maintain my original score.

---

### Author Response · Authors · 2023-11-19
**Response  to all reviewers**

## To all reviewers

Thanks for reviewing our work. As considering the responses of each reviewer, we believe that our manuscript was difficult to understand. To help understand our work, we would like to clarify the targeted problem of our work and explain the proposed method more friendly. Additionally, we reply the common questions of reviewers. For the final manuscript, we are now working to reflect the feedbacks from all reviewers, and will upload it as soon as possible.


## Main motivation of this work

We aim to propose a better function-space prior and corresponding variational distribution for BNNs which incorporate the following principle derived from Bayesian Model Averaging (BMA):

> When each predictive function $f_j(x)$ sampled from the posterior distribution has a peaked probability (low entropy), its averaged prediction $ \frac{1}{J} \sum_{j=1}^{J} f_j (x) $ can exhibit either a peaked or flat probability (low or high entropy) according to the varying levels of inconsistency between predicted classes on $x$.

For instance, consider a scenario where a predictive function $ \\{ f_j \\}_{j=1}^{3} $ is randomly sampled from 3-dimensional probability simplices (3-way classification problem). If an input $x$ closely aligns with the training inputs, we expect each $f_j(x)$ to be certain, for instance, $f_j(x)=[1, 0, 0]$ for $j=1,..,3$, and the resulting averaged predictive probability would be $[1, 0, 0]$. Conversely, if $x$ is far from the training inputs, we may have $f_1(x) = [1, 0, 0], f_2(x)=[0,1,0], f_3(x)=[0,0,1]$, yielding an averaged predictive probability of $[1/3, 1/3, 1/3]$. On the other hand, one can also consider the case where $f_1(x)=f_2(x)=f_3(x)=[1/3, 1/3, 1/3]$, resulting in the same average of $[1/3, 1/3, 1/3]$. Comparing these two scenarios, in the first scenario, while the averaged probability has the maximum entropy possible, the individual samples from the prior have the smallest entropies. What makes the average prediction less confident is the disagreements between the individual samples. On the other hand, for the second scenario, all the samples have the maximum entropies.

What we argue is that a prior favoring the second scenario (e.g., uniform prior on function space) is suboptimal, as it directly encourages the individual samples to have high entropy, and this might lead to less certain predictions for in-distribution data. **In contrast, our proposal encourages the first scenario, where the overall uncertainty in averaged predictions are mainly driven by the disagreement between the individual samples, and the individual samples are not particularly driven to have high entropy.**


**To achieve the explained BMA property**, we refine the function-space prior and variational distribution as follows:


> **Function-space prior:** We refine the function-space prior distribution to have a peaked probability instead of a flat probability.  Using the above example, we expect the sample function of the prior distribution to be $f_1(x)=[0.9, 0.1, 0], \ f_2(x)=[0, 0.8, 0.2], f_3(x)=[0.3,0,0.7]$, similar to $ [1, 0, 0],  [0,1, 0], [0,0,1]$, adaptively depending on how close an input $x$  is to the training dataset (in-distribution dataset), instead of using $ f_1(x)=f_2(x)=f_3(x)=[1/3,1/3, 1/3] $ for any input $x$. To achieve this, we use the weight parameters and penultimate hidden features obtained in the middle stage of training, and then build a random function that outputs a more peaked probability as the input gets closer to the in-distribution dataset. This prior distribution induces the sample function of the variational distribution to have a peaked probability through the KL divergence regularization on function-space.

> **Function-space variational distribution:** We refine the function-space variational distribution to sample predictive functions with varying levels of disagreement between predicted classes  Using the above example, we expect the sample functions of the variational distribution to be $f_j(x)=[1, 0, 0], \(j=1, 2, 3\)$ for an input from the in-distribution dataset and $f_1(x)=[1, 0, 0], f_2(x)=[0, 1, 0], f_3(x)=[0, 0, 1]$ for an input $x$ from the out-of-distribution dataset. To achieve this, we introduce a latent variable that controls the degree of the disagreement for the predicted class of the sample functions, depending on how close an input $x$  is to the in-distribution dataset.

---

> ### Author Response · Authors · 2023-11-19
> **Response  to all reviewers for ""Understanding""**
>
> ## The distinct property of the function-Space prior as input-dependent prior
>
> A BNN can be understood as consisting of two stochastic Neural Networks (NNs): one utilizes the prior distribution of weight parameters, and the other employs the variational distribution of weight parameters. In this context, the function-space prior represents the distribution of the (linearized) stochastic NN's output using the prior distribution of weight parameters. The function-space prior can be used to regularize the stochastic NN (variational distribution) by minimizing the KL divergence between these stochastic NN output distributions. To this end, the function-space prior should specify the auxiliary inputs to be used for constructing the function-space prior (input specification).
>
> This is a distinctive property of the function-space prior compared to the weight-space prior because **the function-space prior** can regularize the stochastic NN (variational distribution) by employing **"input-dependent” prior** distributions on the function-space. In contrast, **the weight-space prior** for the conventional BNN regularizes the stochastic NN (variational distribution) by using **“input-independent” prior** distributions on weight-space.
>
> ## Issue of the existing function-Space prior
>
> The function-space BNN constructs a function-space prior by assigning a zero-mean Gaussian distribution as the weight prior distribution because it was not well understood how the choice of auxiliary input and weight prior distribution affects the corresponding function-space prior distribution. However, this approach results in the input-independent zero-mean function-space prior. This can introduce unnecessary uncertainty if the auxiliary input is similar to the in-distribution dataset, as illustrated in Figure 1 (a). To understand this, **let us consider the training set as CIFAR 10 and the auxiliary inputs as CIFAR 100 where some inputs in CIFAR 100 share the same labels with CIFAR 10. Regularizing the zero-mean function-space priors of these auxiliary inputs (specific subset of CIFAR 100) into the BNN would introduce unnecessary uncertainty.**
>
> ## Direction of the proposed approach
>
> To establish a reasonable function-space prior, we are exploring how the predictive probability, sampled from the posterior distribution, can exhibit appropriate uncertainty based on the similarity of auxiliary inputs to the training set (for example, auxiliary inputs from CIFAR 100 sharing labels with CIFAR 10). Consequently, we have observed that even if each predictive sample function has a peaked probability, the averaged prediction can be uncertain if the predicted classes exhibit varying levels of disagreement. This principle can elucidate why Deep ensemble [1] achieves good performance. Our goal is to integrate this insight into both the function-space prior and the corresponding function-space posterior distribution.
>
> ## Elaboration of the proposed algorithm
>
> To achieve this, we propose a learning algorithm based on two inductive biases commonly used in deep learning methods:
> > **SWAG [2]:** The empirical mean and covariance of weight parameters has been known to play a similar role with the approximate posterior distribution of BNNs weight parameters. We utilize the empirical mean and covariance, obtained during pre-defined iterations, to construct the function-space prior.
>
> > **MAHALANOBIS [3]:** The Mahalanobis distance of the penultimate hidden feature $h(x)$ has known to be effective in discriminating whether a input $x$ belongs to the in-distribution dataset. We use this inductive bias to enable the predictive sample functions of $x$ to have varying degree of disagreement in their predicted class.
>
> ## Procedure of the proposed algorithm
>
> The proposed learning algorithm consists of two phases. Let $T=200$ be a total of training iterations, and $\mathcal{T}=\\{145, 150, 155, 160\\}$ be the pre-defined iterations as following the notation in our manuscript.
>
> > **Initial Phase (0-160)**: Following the methodology similar to [1], we train the mean parameters of the variational distribution of the BNN weight parameters using MAP inference. During this phase, we compute the empirical parameters in Eq. (9) at the pre-defined $ \mathcal{T}=\\{145, 150, 155, 160\\}$ to construct the function-space prior of Eq. (11).
>
> > **Remaining Phase (161-200)**: We train the mean and variance parameters of the BNNs' variational distribution using function-space variational inference with the proposed function-space prior of Eq. (11) and variational distribution of Eq. (15).
>
>
>
>
>
>
>
> [1] Simple and Scalable Predictive Uncertainty Estimation using Deep Ensembles - NeurIPS 17
>
> [2] A simple baseline for bayesian uncertainty in deep learning - NeurIPS 19
>
> [3] A simple unified framework for detecting out-of-distribution samples and adversarial attacks - NeurIPS 18

---

> ### Author Response · Authors · 2023-11-19
> **Response to all reviewers for ""Understanding""**
>
> ## How the proposed methods achieve the BMA property using the inductive bias.
>
> **Role of the function-space prior:**
> >  The proposed function-space prior is designed to induce a more peaked probability distribution (low-entropy) when the auxiliary input is more similar to the in-distribution dataset. Specifically, the proposed function-space prior $f_{BMA}(\cdot) =  \Theta^{L} \ proj(h(\cdot))$ in Eq. (10) is constructed using the averaged weight parameters and penultimate hidden features in Eq. (9), which are obtained from pre-defined iterations ($\mathcal{T}=\\{145, 150, 155, 160\\}$ in the above example). We would like to emphasize that these averaged weight parameters and hidden features can form the approximate posterior based on the fact that the averaged weight parameters using the finite weight samples $\\{w(t)\\}_{t=1}^{T}$, ($T=200$) during training iterations can serve as an approximate posterior of the BNN in **SWAG [2]**.
>
> > In this context, when the auxiliary input $x$ is more similar to the in-distribution dataset, the corresponding $proj(h(x))$ is likely to be one of the penultimate hidden features in Eq. (9), such as ${\hat{m}^{q}_{h}}$ representing the $q$-the label. Then, the sample function of the prior distribution would have a peaked predictive probability for the $q$-th label using the random weight $\Theta^{L}$ modeled by  the averaged weight parameters in Eq. (9) .
>
> **Role of the function-space prior:**
> >  The proposed function-space variational distribution $q(f)$  is designed to allow the sample functions $ \\{ f_{j} (x) \\}$ of the variational distribution to exhibit varying levels of disagreement between predicted classes on $x$. Specifically, for an input $x$, we model the level of disagreement by introducing the categorical latent variable $z(x) \sim Cat(\hat{p}(x))$ parameterized by $\hat{p}(x) = \frac{ \mu^{L}_{w_q}   \ h(x) }{ \sqrt{ 1 +  \pi/8 || \Delta  \ h(x)  ||} } \in \Delta^{Q-1} $ in Eq. (13).
>
> > We would like to emphasize that $|| \Delta  \ h(x)  || \in R^{Q}$ in the denominator denotes the $Q$-dimensional mahalanobis distance of  $h(x)$ from the averaged penultimate hidden features $\hat{m}^{q}_{h}$ for $q=1,..,Q$ in Eq. (9). The $|| \Delta  \ h(x)  ||$ has known to be effective for OOD detection in **MAHALANOBIS [3]** because its has the small distance for the specific $q$-th dimension or has the large distance for all $q=1,..,Q$, depending on how close a input $x$ is to the in-distribution.
>
> > In this context, when the $x$ seems to be a out-of-distribution dataset, the $\hat{p}(x)$ would become a flat probability due to $|| \Delta  \ h(x)  || $ . This induces the latent variable $z(x)$ to have a different level of disagreement depending on $x$, and thus affects the sample functions $f (x)$ to exhibit varying levels of disagreement by applying the temperature parameter $\tau$ on the specific dimension modeled by $z(x)$.

---

> ### Author Response · Authors · 2023-11-19
> **Response to all reviewers for ""Question on Last-Layer approximation""**
>
> ## Reasons for taking the Last-layer BNN.
>
> **Memory Issue:**
>
> > For widely-used deep neural networks (DNNs) like ResNet, computing the Jacobian matrix for the entire weight parameters demands a substantial amount of GPU memory due to the large size of the Jacobian matrix, which is \#batch size x \# weight parameters x #class; the size of Jacobian matrix for the ResNet 18 on CIFAR 10 using 128 batch is 128 x 11.8M  x 10. This can potentially lead to out-of-memory issues, making it impractical to consider linearization for all weight parameters.
>
> >  Indeed, due to this potential issue, the function-space prior in [4] employs the Jacobian matrix of the last layer for the covariance of the function-space prior. This approach motivates us; if the last layer is modeled as the Bayesian layer allowing the closed form of the function-space distribution, the function-space prior can be constructed even with limited GPU resources. The sample function of the variational distribution can also be efficiently constructed, as described in A.2 in the Appendix (page 12).
>
> **Effectiveness of the Last-layer BNN:**
>
> >  In large-scale BNNs, considering partially stochastic layers for BNNs can be more effective than using fully stochastic layers [5]. Based on this observation, we designate the last layer as the Bayesian layer. To address the necessary uncertainty modeled by the $L-1$ stochastic layers $\\{f^{l} \\}_{l=1}^{L-1}$, we directly model this uncertainty through the latent variable $z(x)$, by leveraging the $Q$-dimensional mahalanobis distance $|| \Delta  \ h(x)  ||$ for the penultimate feature $h(x)$ from using the averaged penultimate features in Eq. (9).
>
> ## Marginal Performance improvement, possibly due to the Last-layer BNN.
>
> To investigate the effectiveness of the last-layer approximation, we conducted additional experiments on CIFAR-10 using ResNet-18 as the baseline model. Due to the memory issues explained above, we extended the Bayesian last layer for the baseline (T-FVI) by replacing the last linear layer ([512, 10] with 512 layer features and 10 classes) with a Bayesian 2-hidden MLP layer ([512, 128] -> ReLU -> [128, 10]). Note that this Bayesian 2-hidden MLP layer uses an increasing number of weight parameters for the mean and variance parameters of the variational and prior distributions.
>
> Additionally, we considered varying numbers of training sets, specifically $\\{10,000, 20,000, 30,000, 45,000\\}$, to demonstrate the effectiveness of the proposed inference. For the hyperparameters, we tuned the regularization hyperparameter $\lambda$ in $\\{10^{-3}, 10^{-4}, 10^{-5}\\}$ for KL divergence and $r$ in $\\{0.05, 0.10\\}$ for the adversarial hidden feature with the temperature parameter $\tau=1/0.9$ for the variational distribution.
>
> We report ACC, NLL, AUROC, and ECE, respectively, following the protocol in the manuscript. We observe that R-FVI consistently outperforms the baselines in terms of NLL and AUROC, except when using a $10,000$ training set. We would like to emphasize that the improvement of R-FVI is not marginal when comparing its performance with that of T-FVI (2 MLP Layers).
>
> | ACC       | R-FVI               | T-FVI               | T-FVI (2 MLP Layers) |
> |--------------|---------------------|---------------------|----------------------|
> | 45,000 (Full)         | (**0.951**, 0.000)      | (0.949, 0.002)      | (0.948, 0.001)       |
> | 30,000       | (**0.939**, 0.001)      | (0.938, 0.002)      | (0.935, 0.002)       |
> | 20,000       | (**0.923**, 0.001)      | (0.920, 0.002)      | (0.917, 0.002)       |
> | 10,000       | (**0.880**, 0.001)      | (0.877, 0.001)      | (0.873, 0.003)       |
>
> | NLL       | R-FVI               | T-FVI               | T-FVI (2 MLP Layers) |
> |--------------|---------------------|---------------------|----------------------|
> |45,000 (Full)        | (**0.167**, 0.001)      | (0.176, 0.008)      | (0.175, 0.002)       |
> | 30,000       | (**0.207**, 0.001)      | (0.214, 0.004)      | (0.215, 0.004)       |
> | 20,000       | (**0.260**, 0.004)      | (0.270, 0.005)      | (0.274, 0.001)       |
> | 10,000       | (**0.400**, 0.010)      | (0.402, 0.006)      | (0.411, 0.006)       |
>
> | AUROC       | R-FVI               | T-FVI               | T-FVI (2 MLP Layers) |
> |--------------|---------------------|---------------------|----------------------|
> | 45,000 (Full)         | (**0.957**, 0.002)      | (0.938, 0.012)      | (0.932, 0.012)       |
> | 30,000       | (**0.908**, 0.019)      | (0.891, 0.025)      | (0.906, 0.021)       |
> | 20,000       | (**0.912**, 0.016)      | (0.904, 0.009)      | (0.887, 0.017)       |
> | 10,000       | (0.794, 0.03)       | (0.801, 0.039)      | (**0.838**, 0.035)       |

---

> ### Author Response · Authors · 2023-11-19
> **Response to all reviewers for ""Question on Last-Layer approximation""**
>
> | ECE       | R-FVI               | T-FVI               | T-FVI (2 MLP Layers) |
> |--------------|---------------------|---------------------|----------------------|
> | 45,000 (Full)           | (0.010, 0.001)      | (0.012, 0.001)      | (0.010, 0.001)       |
> | 30,000       | (0.012, 0.0)        | (0.011, 0.0)        | (0.012, 0.002)       |
> | 20,000       | (0.014, 0.0)        | (0.014, 0.0)        | (0.013, 0.0)         |
> | 10,000       | (0.017, 0.002)      | (0.016, 0.001)      | (0.017, 0.001)       |
>
> [4] Tractable Function-Space Variational Inference in Bayesian Neural Networks - NeurIPS 22
>
> [5] Do Bayesian Neural Networks Need To Be Fully Stochastic? - AISTATS 23

---

> ### Author Response · Authors · 2023-11-19
> **Response to all reviewers for ""Extension for regression tasks""**
>
> ## Modification for Regression
>
> For regression task, the proposed method can be also used with a minor modification as the following:
>
> > For a continuous-valued output $Y=\\{y_{n}\\}_{n=1}^{N}$ in the training set, we consider the range of $[-\infty, \min{Y}] \cup [\min{Y}, \max{Y}] \cup [\max{Y}, \infty]$, partitioning this range into $Q$ intervals, including $(-\infty, \min{Y})$ and $(\max{Y}, \infty)$. We assign the order of each interval as the discrete label. For instance, if $y_n$ belongs to the $q$-th interval out of $Q$ intervals ($1 \leq q \leq Q$), we define the discrete label of $y_n$ as $q$. Then, we compute Eq. (8) using the assigned discrete label for each $y_n$, building the function-space prior in Eq. (10) as done in the classification task.
>
> > Regarding the variational distribution, we consider the latent variable $z(x) \sim Cat(  \hat{p}(x)  )$ using mahalanobis distance, i.e., $\hat{p}(x)  = || \Delta h(x) ||$ for an input $x$. Then, we apply the parameter $\tau$ to the only covariance parameter $\Sigma_{f|z}$ in Eq. (15), enabling the sample function $f(x)$ to exhibit varying levels of predictive uncertainty depending on the interval corresponding $x$. However, we believe that covering this approach in this work is too extensive. Therefore, for the regression, we recommend using the proposed function-space prior without using the latent variable $z$. Indeed, we confirm that using the regularization by the proposed function-space prior yields good performance, as below:
>
>
> ## Experiment Result
>
> We validate the proposed method (R-FVI) on the UCI regression task and compare R-FVI with FVI [6] and T-FVI [4] using the GP prior with the RBF kernel. We follow the same setting as the UCI regression experiment in the appendix of [4]. For the proposed method, we set $Q=20$ intervals and find the best regularization hyperparameter $\lambda \in \\{0.001, 0.01, 0.1\\}$ for KL divergence and the radius $r \in \\{0.01, 0.1 \\}$ in Eq. (12). We apply map inference for the first 50 percent of the total training iterations and then apply function-space variational inference for the remaining iterations.
>
> The first table reports RMSE ($\downarrow$: better), and the second table reports log likelihood, ($\uparrow$: better). We observe that the proposed method tends to outperform the baselines FVI and T-FVI.
>
> | RMSE    | FVI              | T-FVI            | R-FVI (our)      |
> |------------|------------------|------------------|------------------|
> | **Boston**     | (2.328, 0.104)   | (3.632, 0.515)   | (**2.271**, 0.431)   |
> | **Concrete**   | (4.935, 0.180)   | (4.177, 0.443)   | (**4.005**, 0.461)   |
> | **Energy**     | (0.412, 0.017)   | (0.409, 0.060)   | (**0.334**, 0.040)   |
> | **Wine**       | (0.673, 0.014)   | (0.615, 0.033)   | (**0.495**, 0.032)   |
> | **Yacht**      | (0.607, 0.068)   | (0.514, 0.242)   | (**0.316**, 0.099)   |
> | **Protein**    | (4.326, 0.019)   | (4.248, 0.043)   | (**3.870**, 0.027)   |
>
> | Log likelihood (LL)    | FVI               | T-FVI             | R-FVI (our)       |
> |------------|-------------------|-------------------|-------------------|
> | **Boston**     | (-2.301, 0.038)   | (-3.150, 0.495)   | (**-2.173**, 0.520)  |
> | **Concrete**   | (-3.096, 0.016)   | (-2.855, 0.116)   | (**-2.371**, 0.256)   |
> | **Energy**     | (-0.684, 0.020)   | (-0.539, 0.138)   | (**-0.346**, 0.165)   |
> | **Wine**       | (-1.040, 0.013)   | (-0.959, 0.034)   | (**-0.063**, 0.084)   |
> | **Yacht**      | (-1.033, 0.033)   | (-0.888, 0.334)   | (**-0.289**, 1.383)   |
> | **Protein**    | (-2.892, 0.004)   | (-2.866, 0.009)   | (**-2.079**, 0.004)   |
>
>
> ### Sensitivity analysis for $Q$ intervals
>
> Furthermore, we conducted additional experiments to assess the robustness of the proposed method across different $Q$ interval on the ''Concrete,' 'Energy,' and 'Protein' datasets. The results, presented below, illustrate the consistent performance of the method under varying number of interval $Q \in \\{5, 10, 15, 20 \\}$.
>
>
> | RMSE   | 5                | 10               | 15               | 20               |
> |--------|------------------|------------------|------------------|------------------|
> | **Concrete** | (3.724, 0.437)   | (3.727, 0.428)   | (3.628, 0.508)   | (4.005, 0.461)   |
> | **Wine**    | (0.493, 0.025)   | (0.496, 0.028)   | (0.496, 0.029)   | (0.495, 0.032)   |
> | **Protein** | (3.728, 0.023)   | (3.766, 0.041)   | (3.726, 0.024)   | (3.870, 0.027)   |
>
> | Log likelihood (LL)     | 5                | 10               | 15               | 20               |
> |--------|------------------|------------------|------------------|------------------|
> | **Concrete** | (-2.306, 0.231)  | (-2.471, 0.391)  | (-2.400, 0.343)  | (-2.371, 0.256)  |
> | **Wine**    | (-0.080, 0.075)  | (-0.085, 0.089)  | (-0.059, 0.085)  | (-0.063, 0.084)  |
> | **Protein** | (-2.034, 0.004)  | (-2.04, 0.006)   | (-2.034, 0.004)  | (-2.079, 0.004)  |
>
> [6] Functional Variational Bayesian Neural Networks - ICLR 19

---

### Author Response · Authors · 2023-11-23
**Additional Responses to all reviewers**

We appreciate your additional responses to our replies. Through this rebuttal process, we've gained a clearer understanding of the identified weaknesses in our research and manuscript, especially in terms of readability, methodology explanation for the proposed function-space prior construction, regression extension, and the elaboration of the last-layer approximation.

To address these issues, we commit to making substantial revisions to our manuscript, incorporating the feedback and suggestions provided during this process. However, due to the limited time relative to the required changes, we will maintain the manuscript in its current state for this round. We appreciate your constructive feedback once again.

---

### Meta-Review · Area_Chair_HKKU · 2023-12-08

**Metareview:**

This paper discusses ideas to refine Bayesian neural networks from a function space view. This a very active and interesting topic, and the reviewers appreciate the basic approach presented. Specifically, the paper aims at constructing a more interpretable prior in function space.

However, the exposition generally lacks in clarity and key parts of the technical contribution are not clear. It is questionable how the proposed techniques are supposed to implement the presented ideas. Furthermore, the reviewers criticized the limited experimental evaluation, rendering the efficacy of the approach questionable.

**Justification For Why Not Higher Score:**

Weaknesses above: unclear presentation and paper writing, limited experiments.

**Justification For Why Not Lower Score:**

NA

---

### Decision · Program_Chairs · 2024-01-16

Reject